# RETHINKING THE "HEATMAP + MONTE CARLO TREE SEARCH" PARADIGM FOR SOLVING LARGE SCALE TSP

## ABSTRACT

The Travelling Salesman Problem (TSP) remains a fundamental challenge in combinatorial optimization, inspiring diverse algorithmic strategies. This paper revisits the "heatmap + Monte Carlo Tree Search (MCTS)" paradigm that has recently gained traction for learning-based TSP solutions. Within this framework, heatmaps encode the likelihood of edges forming part of the optimal tour, and MCTS refines this probabilistic guidance to discover optimal solutions. Contemporary approaches have predominantly emphasized the refinement of heatmap generation through sophisticated learning models, inadvertently sidelining the critical role of MCTS. Our extensive empirical analysis reveals two pivotal insights: **1**) The configuration of MCTS strategies profoundly influences the solution quality, demanding meticulous tuning to leverage their full potential; **2**) Our findings demonstrate that a rudimentary and parameter-free heatmap, derived from the intrinsic $k$-nearest nature of TSP, can rival or even surpass the performance of complicated heatmaps, with strong generalizability across various scales. Empirical evaluations across various TSP scales underscore the efficacy of our approach, achieving competitive results. These observations challenge the prevailing focus on heatmap sophistication, advocating a reevaluation of the paradigm to harness both components synergistically.

## 1 INTRODUCTION

The Travelling Salesman Problem (TSP) stands as a quintessential challenge in combinatorial optimization, drawing considerable interest from both theoretical and applied research communities. As a problem characterized by NP-hardness, the TSP has become a benchmark for evaluating the efficacy of novel algorithmic strategies in determining optimal or near-optimal solutions efficiently (Applegate et al., 2009). It has significant practical applications in domains such as logistics, transportation, manufacturing, and telecommunications, where finding efficient routes is crucial for minimizing costs and improving efficiency (Helsgaun, 2017; Nagata & Kobayashi, 2013).

Recent advancements in machine learning have inspired a fresh wave of methodologies for tackling the TSP, particularly through the lens of the "heatmap + Monte Carlo Tree Search (MCTS)" paradigm. This innovative approach, first introduced by Fu et al. (2021), leverages learning mechanisms to approximate edge probabilities in forming part of the optimal path (heatmap), while employing MCTS to intelligently search and refine this probabilistic scaffold. The method successfully solved TSP instances with 10,000 nodes, inspiring more researchers to contribute to this solving paradigm (Qiu et al., 2022; Sun & Yang, 2023; Min et al., 2024).

Historically, the focus within this paradigm has predominantly centered on the design and refinement of effective heatmaps, as these serve as the foundational guides for MCTS. Sophisticated learning-based designs, ranging from supervised learning (Fu et al., 2021) to diffusion models (Sun & Yang, 2023), are employed to predict these probabilities with high accuracy, thus assuming that the sophistication of the heatmap directly correlates with solution quality. However, this singular emphasis may have inadvertently overshadowed the critical contribution of the MCTS phase. The MCTS stage, tasked with exploring the solution space given the probabilistic cues from the heatmap,

has the potential to significantly refine or degrade the final solution quality depending on its strategy configuration (Xia et al., 2024).

In this study, we rigorously examine the underexplored dimension of the MCTS strategy within the heatmap-guided paradigm, challenging the prevailing narrative that sophisticated heatmaps are the singular key to superior solutions. We demonstrate that with carefully calibrated MCTS strategies, the efficacy of the solution can be markedly enhanced, spotlighting the need for a dual focus on both components of the paradigm. While SoftDist (Xia et al., 2024) also noticed that the competitive performance can be obtained through a simple parameterization of heatmaps, we move forward arguing that a parameter-free heatmap deriving from the $k$-nearest generalizable prior can achieve better. Our findings not only uncover the potential of such naïve methods, but also provide novel perspectives to rethink essential factors in methodical designs for TSP.

Overall, our contributions are threefold:

- We elucidate the substantial role of MCTS configurations in optimizing TSP solutions, encouraging a reevaluation of existing priorities in algorithm design. We demonstrate that fine-tuning MCTS parameters such as exploration constant and node expansion criteria can significantly impact solution quality.

- We demonstrate that simplicity in heatmap construction, based on $k$-nearest statistics and devoid of parameters and training, does not necessarily equate to inferior results. Furthermore, it exhibits scalability across various sizes, thereby expanding the possibilities for methodological innovations.

- We present empirical evidence of our approach achieving competitive performance across large TSP scales.

These insights collectively advocate for a more balanced integration of learning and search, potentially guiding future research endeavors towards more holistic algorithmic frameworks that better capitalize on the inherent symbiosis of these components.

## 2 RELATED WORKS

The integration of machine learning techniques with combinatorial optimization has led to significant advancements in solving large-scale TSPs. We focus on approaches that have demonstrated capability in addressing TSP instances with tens of thousands of nodes, particularly those employing the heatmap-guided MCTS paradigm. For additional details on the neural solver for addressing TSP, please see the Appendix A.

Fu et al. (2021) introduced a pioneering framework combining graph convolutional networks with attention mechanisms to generate edge probability heatmaps for TSP. This approach guides MCTS to refine complete tours instead of constructing partial solutions, marking a significant departure from traditional MCTS methods. The framework's efficacy in handling large-scale problems has established it as a foundational approach for subsequent learning-based TSP solvers with heatmap-guided MCTS.

Building upon this foundation, several methods have emerged, focusing predominantly on enhancing heatmap generation through sophisticated learning models. Qiu et al. (2022) proposed DIMES, a differentiable meta-solver employing Graph Neural Networks (GNNs) to parameterize the solution space. Sun & Yang (2023) introduced DIFUSCO, leveraging diffusion models for heatmap generation, while Min et al. (2024) developed UTSP, an unsupervised learning approach to address challenges in data efficiency and reward sparsity. More recently, approaches like SoftDist (Xia et al., 2024) have begun to explore simpler heatmap construction methods, introducing a temperature coefficient to craft heatmaps based on distance. This shift towards simpler heatmaps suggests a growing recognition of the potential overemphasis on heatmap sophistication.

While existing methods have primarily focused on refining heatmap generation, our work addresses a gap in the literature by critically examining the balance between heatmap construction and MCTS optimization. We analyze the effects of MCTS hyperparameters and propose a k-nearest prior heatmap, thereby enhancing our understanding of how heatmap-guided MCTS can be optimized for large-scale TSP solutions

## 3 PRELIMINARY

To establish a solid foundation for our analysis, we define the Travelling Salesman Problem (TSP) and outline the Monte Carlo Tree Search (MCTS) framework as applied to TSP solutions.

### 3.1 PROBLEM DEFINITION

A TSP instance of size $N$ is formulated as a set of points $I = \{(x_i, y_i)\}_{i=1}^N$ in the Euclidean plane, where each point represents a city with coordinates $(x_i, y_i) \in [0, 1] \times [0, 1]$. The distance $d_{ij}$ between any two cities $i$ and $j$ is determined by the euclidean distance formula, defined as: $d_{ij} = \sqrt{(x_i - x_j)^2 + (y_i - y_j)^2}$. The goal is to find the shortest possible route that visits each city exactly once and returns to the origin city. This optimal route is represented by the permutation $\pi^* = (\pi_1^*, \pi_2^*, \dots, \pi_N^*)$ of the sequence $(1, 2, \dots, N)$, with the shortest tour length: $L(\pi^*) = \sum_{i=1}^{N-1} d_{\pi_i^* \pi_{i+1}^*} + d_{\pi_N^* \pi_1^*}$. We can determine the optimality gap of a feasible tour $\pi$ by

$$Gap = \left( \frac{L(\pi)}{L(\pi^*)} - 1 \right) \times 100\%. \tag{1}$$

In the "Heatmap + MCTS" paradigm, a central concept is the heatmap, depicted as an $N \times N$ matrix $P^N$. Each element $P_{ij}^N \in [0, 1]$ denotes the probability of edge $(i, j)$ being part of the optimal TSP solution, providing a probabilistic guide for the search process.

### 3.2 MONTE CARLO TREE SEARCH FRAMEWORK

The MCTS framework is modeled as a Markov Decision Process (MDP), which is represented by states, actions and the transition between states. The implementation is built upon the framework proposed by Fu et al. (2021), integrating learned heatmaps to enhance search efficiency.

In this framework, each state $\pi$ represents a feasible TSP tour, a permutation of the index of cities. The initial state is constructed by iteratively selecting edges with a probability proportional to $e^{P_{ij}^N}$. Actions are defined as $k$-opt moves, which modify the current tour by replacing $k$ edges to create a new tour. The metric of a state $\pi$ is defined as the tour length $L(\pi)$.

Algorithmically, the MCTS for solving TSP consists of four primary steps:

1. **Initialization**: The weight matrix $W$ is computed from the heatmap matrix $P^N$, with each element defined as $W_{ij} = 100 \times P_{ij}^N$, representing the probability of selecting edge $(i, j)$. This approach adheres to the method introduced by Fu et al. (2021). The access matrix $Q$ is initialized with all elements set to zero ($Q_{ij} = 0$) to track the frequency with which each edge is selected, while $M$ is initialized to zero to enumerate the total number of actions. Additionally, a candidate set is constructed for each node, and subsequent edges are exclusively selected from this candidate set.

2. **Simulation**: A number of actions (candidate $k$-opt moves) are generated by selecting edges based on current state. The selection probability of an edge is proportional to its potential, calculated as:

$$Z_{ij} = \frac{W_{ij}}{\Omega_i} + \alpha \sqrt{\frac{\ln(M+1)}{Q_{ij}+1}}, \tag{2}$$

where $W_{ij}$ is the edge weight, $\Omega_i = \sum_{j \neq i} W_{ij}$ normalizes for node $i$, $Q_{ij}$ tracks edge usage, and $\alpha$ controls the exploration-exploitation tradeoff.

3. **Selection**: During the simulation, if an improving action from the sampling pool is found, it will be accepted to convert current state $\pi$ into $\pi'$ with $\Delta L = L(\pi') - L(\pi) < 0$. Otherwise, the method jumps to a random state by initializing another tour based on $P^N$, which becomes the new starting point for exploration.

4. **Back-propagation**: After applying an action, the weight matrix $W$ is updated to reflect the improvement in the tour length:

$$W_{ij} \leftarrow W_{ij} + \beta \left( \exp \left( \frac{L(\pi) - L(\pi')}{L(\pi)} \right) - 1 \right) \tag{3}$$

where $\beta$ is the learning rate. This update process promotes edges leading to better tours. The access matrix $Q$ is also incremented for edges involved in the action.

**Termination:** The MCTS process continuously repeats the four steps until a predefined termination criterion (time limit) is met. The best state found is returned as the final solution.

# 4 THE IMPORTANCE OF HYPERPARAMETER TUNING

In this section, we highlight the often-overlooked importance of properly configuring MCTS hyperparameters, a factor crucial for improving solution quality, while recent studies have primarily focused on advanced heatmap generation.

## 4.1 OVERVIEW OF HYPERPARAMETERS

We have identified several key hyperparameters that significantly influence the performance of the heatmap-guided MCTS approach:

`Alpha`: Controls the exploration-exploitation balance in MCTS by weighting the exploration term in Eq. (2). Higher values promote exploration of unvisited nodes, while lower values favor exploitation of known good paths.

`Beta`: Influences the MCTS Back-propagation process as defined in Eq. (3). Higher values lead to more aggressive updates, potentially causing rapid shifts in search strategy based on recent results.

`Max_Depth`: Sets the maximum depth for $k$-opt moves during MCTS simulation. Larger values allow more complex moves at the cost of increased computation time.

`Max_Candidate_Num`: Limits the candidate set size at each node, sparsifying the graph and affecting both algorithm speed and solution quality. Smaller sets accelerate search but may overlook optimal solutions.

`Param_H`: Determines the number of simulation attempts per move, with a maximum of `Param_H`$\times$ $N$ simulations. Higher values provide more comprehensive exploration at the expense of increased computation time.

`Use_Heatmap`: A boolean parameter that decides whether candidate set construction is guided by heatmap probabilities or distances. When enabled, it can enhance search efficiency if heatmaps are accurate.

`Time_Limit`: Sets the overall search time limit to `Time_Limit` $\times N$ seconds. MCTS terminates and returns a solution upon reaching this limit. The default value is 0.1, as specified in Fu et al. (2021).

These hyperparameters collectively influence the MCTS process. In our tuning experiments, we optimize all parameters except `Time_Limit`, which remains fixed unless otherwise specified.

## 4.2 HYPERPARAMETER SEARCH AND IMPORTANCE ANALYSIS

To determine the impact of different hyperparameters, we conducted a comprehensive hyperparameter search on the TSP-500, TSP-1000 and TSP-10000 training sets. Datapoints were generated following Fu et al. (2021). We applied learning-based methods (Att-GCN (Fu et al., 2021), DIMES (Qiu et al., 2022), DIFUSCO (Sun & Yang, 2023), UTSP (Min et al., 2024), SoftDist (Xia et al., 2024)), non-learnable Zero baseline, and our proposed GT-Prior method (detailed in Section 5.1) to generate heatmaps for grid search. The initial search space (Table 1) was based on MCTS settings from Att-GCN and UTSP, and algorithm dynamics analysis. Bold configurations represent default settings from previous works (Fu et al., 2021; Qiu et al., 2022; Sun & Yang, 2023; Xia et al., 2024). Post grid search, we employed the S̲Hapley A̲dditive exP̲lanations (SHAP) method (Lundberg & Lee, 2017; Lundberg et al., 2020) to analyze hyperparameter importance. SHAP is a game theory-based approach that assigns importance values to features based on their contributions

---

In this context, the training set refers to the data used for hyperparameter tuning, distinct from the set used for model training.

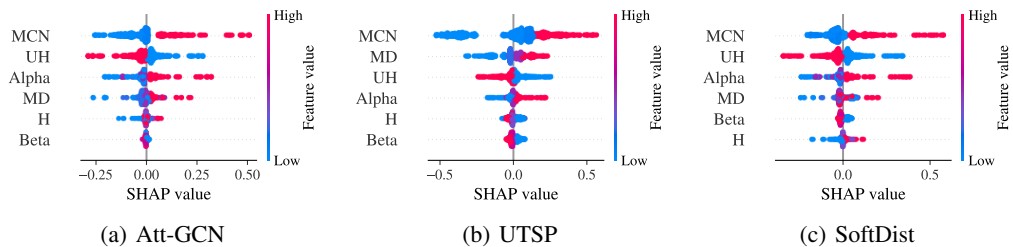

(a) Att-GCN        (b) UTSP        (c) SoftDist

Figure 1: Beeswarm plots of SHAP values for three different heatmaps. MD: `Max_Depth`, MCN: `Max_Candidate_Num`, H: `Param_H`, UH: `Use_Heatmap`. Each dot represents a feature's SHAP value for one instance, indicating its impact on the TSP solution length. The x-axis shows SHAP value magnitude and direction, while the y-axis lists features. Vertical stacking indicates similar impacts across instances. Wider spreads suggest greater influence and potential nonlinear effects. Dot color represents the corresponding feature value.

to a model's output. In our context, SHAP values indicate each hyperparameter's impact on TSP solution quality. Positive values suggest increased solution length (worse performance), while negative values indicate reduced length (better performance).

Figure 1 presents SHAP value distributions for hyperparameters across three heatmap models (Att-GCN, UTSP, and SoftDist) on TSP-500, with additional plots for different models and problem sizes in Appendix C. `Max_Candidate_Num` consistently shows a strong, often positive impact across models, suggesting that reducing the candidate set improves both speed and solution quality. `Max_Depth` generally exhibits positive SHAP values, indicating that deeper explorations tend to worsen performance.

Table 1: The search space. The bolded configuration indicates the default settings.

| Hyperparameter | Range |
|---|---|
| `Alpha` | $[0, \mathbf{1}, 2]$ |
| `Beta` | $[\mathbf{10}, 100, 150]$ |
| `Max_Depth` | $[\mathbf{10}, 50, 100, 200]$ |
| `Max_Candidate_Num` | $[5, 20, 50, \mathbf{1000}]$ |
| `Param_H` | $[2, 5, \mathbf{10}]$ |
| `Use_Heatmap` | $[\mathbf{True}, False]$ |

`Alpha` and `Use_Heatmap` display mixed effects, revealing non-linear interactions where their impact varies depending on the heatmap. `Beta` shows a strong positive influence in SoftDist, implying that suboptimal update strategies negatively affect its performance. `Param_H` demonstrates minimal overall influence across the examined heatmaps.

## 4.3 PERFORMANCE IMPROVEMENT THROUGH HYPERPARAMETER TUNING

**Experimental Setup** The hyperparameter tuning process involves three different problem scales: TSP-500, TSP-1000, and TSP-10000. The training set used for tuning consisted of 128 instances each for TSP-500 and TSP-1000, and 16 instances for TSP-10000. To evaluate the performance of different hyperparameter settings, we utilized the test set provided by Fu et al. (2021). The MCTS computations and grid search were performed on an AMD EPYC 9754 128-Core CPU. During grid search, the `Time_Limit` for MCTS was set to 0.1 for TSP-500 and TSP-1000, 0.01 for TSP-10000. A detailed discussion of the `Time_Limit` is provided in Section 6.1.

**Metrics** We evaluated performance using two metrics: *Gap* defined in Eq. (1), which represents the relative performance gap in solution length compared to a baseline method (Concorde (Applegate et al., 2009) for TSP-500, TSP-1000 and LKH-3 (Helsgaun, 2017) for TSP-10000), and *Improvement*, which refers to the relative reduction in the gap after hyperparameter tuning.

**Baselines** We tuned and evaluated heatmaps generated by seven different methods: Zero , Att-GCN, DIMES, DIFUSCO, UTSP, SoftDist and GT-Prior. To generate heatmaps for the training set, we utilized the code and model checkpoints provided by the corresponding works, while the

We set the `Use_Heatmap` parameter to `False` while maintaining other parameters as specified in Table 1 for the Zero heatmap, since Zero heatmap provides no information about the instances.

Table 2: Performance Improvement after Hyperparameter Tuning.

| METHOD | MCTS SETTING | TSP-500 | | TSP-1000 | | TSP-10000 | |
|---|---|---|---|---|---|---|---|
| | | GAP ↓ | IMPROVEMENT ↑ | GAP ↓ | IMPROVEMENT ↑ | GAP ↓ | IMPROVEMENT ↑ |
| ZERO | DEFAULT | 3.60% | | 4.70% | | 5.45% | |
| | TUNED | 0.66% | 2.93% | 1.16% | 3.54% | 3.79% | 1.66% |
| ATT-GCN | DEFAULT | 1.47% | | 2.26% | | 3.62% | |
| | TUNED | 0.69% | 0.79% | 1.09% | 1.17% | 3.02% | 0.60% |
| DIMES | DEFAULT | 1.57% | | 2.30% | | 3.05% | |
| | TUNED | 0.69% | 0.89% | 1.11% | 1.19% | 3.85% | -0.79% |
| UTSP | DEFAULT | 3.14% | | 4.20% | | — | |
| | TUNED | 0.90% | 2.24% | 1.53% | 2.67% | — | — |
| SOFTDIST | DEFAULT | 1.22% | | 2.00% | | 2.94% | |
| | TUNED | 0.44% | 0.79% | 0.80% | 1.19% | 3.29% | -0.34% |
| DIFUSCO | DEFAULT | 0.45% | | 1.07% | | 2.69% | |
| | TUNED | 0.33% | 0.12% | 0.53% | 0.54% | 2.36% | 0.32% |
| GT-PRIOR | DEFAULT | 1.41% | | 2.12% | | 3.10% | |
| | TUNED | 0.50% | 0.91% | 0.85% | 1.27% | 2.13% | 0.97% |

heatmaps for the test set were provided by Xia et al. (2024). It's worth noting that UTSP does not provide a way to generate heatmaps for TSP-10000. For simplicity, we utilize grid search to tune the hyperparameters.

**Results**   Table 2 summarizes our hyperparameter tuning experiments, revealing significant improvements in solution quality across all methods. Performance gains were particularly pronounced for heatmaps with modest initial performance, such as UTSP, which improved from a 3.14% gap to 0.90% (a 2.24% reduction) on TSP-500. Even high-performing methods like DIFUSCO showed notable improvements: 0.12% on TSP-500 and 0.54% on TSP-1000. Some methods experienced slight performance drops after tuning, potentially due to differences between tuning and test instances. Detailed post-tuning hyperparameter settings are provided in Appendix F.

The computational effort required for hyperparameter tuning is comparable to the training time of learning-based methods, both in scale and impact on subsequent performance. Our tuning process, conducted via grid search, is a one-time investment that incurs no additional computational costs during inference. The efficiency of this process can be further enhanced through increased parallelization and advanced hyperparameter optimization algorithms such as SMAC3 (Lindauer et al., 2022). For a detailed discussion on tuning performance, efficiency, and comparative results including those from SMAC3, please refer to Appendix G.

These findings highlight the critical importance of hyperparameter tuning in optimizing heatmap-guided MCTS for TSP solving. Our results suggest that a balanced approach, considering both heatmap design and MCTS optimization, can yield superior outcomes compared to focusing solely on heatmap sophistication. Notably, even with simpler heatmap construction methods (Zero and GT-Prior), one can achieve competitive performance when coupled with carefully tuned MCTS, rivaling more complex, learning-based approaches. Furthermore, our analysis reveals that the often-overlooked postprocessing of heatmaps has a non-negligible impact on the final TSP solution quality. A detailed examination of heatmap postprocessing effects is provided in Appendix B.

## 5   A PARAMETER-FREE BASELINE BASED ON $k$-NEAREST PRIOR

In the realm of heatmap-guided MCTS for TSP solving, the construction of effective heatmaps has predominantly relied on complex learning methods (Fu et al., 2021; Qiu et al., 2022; Sun & Yang, 2023; Min et al., 2024) or parameterized approaches (Xia et al., 2024). These methods, while often effective, can be computationally intensive during training or testing and may lack generalizability across different problem scales. In this section, we introduce a simple yet effective baseline method that capitalizes on the $k$-nearest prior commonly observed in optimal TSP solutions (see Section 5.1). This approach eliminates the need for parameter tuning, showcasing robust perfor-

mance (see Section 5.2) and strong generalization capabilities across different problem sizes (see Section 5.3).

## 5.1 THE $k$-NEAREST PRIOR IN TSP

The $k$-nearest prior in TSP refers to the observation that in optimal solutions, the next city visited is frequently among the $k$ nearest neighbors of the current city, where $k$ is typically a small value. This property has been implicitly utilized in various TSP solving approaches, including the construction of sparse graph inputs for deep learning architectures (Fu et al., 2021; Sun & Yang, 2023; Min et al., 2024). However, the statistical characteristics and optimal selection of $k$ have been underexplored in the literature.

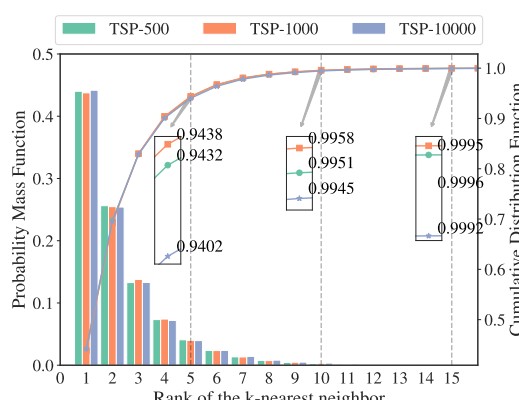

Figure 2: Empirical distribution of $k$-nearest neighbor selection in optimal TSP tours

To elucidate the $k$-nearest prior, we conducted a comprehensive analysis of (near-) optimal solutions for TSP instances of various sizes. Given a set of TSP instances $\mathcal{I}$, for each instance $I \in \mathcal{I}$ and its optimal solution, we calculate the rank of the nearest neighbors for the next city: $k \in \{1, 2, ..., N\}$, and count their occurrences $n_k^I$, where $n_k^I$ represents the number of selecting the $k$-nearest cities in an instance's optimal solution. We then calculate the distribution:

$$\mathbb{P}_N^I(k) = \frac{n_k^I}{N}, k \in \{1, 2, ..., N\} \tag{4}$$

and average these distributions across all instances to derive the empirical distribution of the $k$-nearest prior:

$$\hat{\mathbb{P}}_N(k) = \frac{1}{|\mathcal{I}|} \sum_{I \in \mathcal{I}} \mathbb{P}_N^I(k), k \in \{1, 2, ..., N\}. \tag{5}$$

To visualize the empirical distribution $\hat{\mathbb{P}}_N(\cdot)$, we first generate instances by uniformly sampling cities from the unit square. Then we examined 3000 optimal solutions for TSP-500 and TSP-1000 using the Concorde solver, as well as 128 near-optimal solutions for TSP-10000 employing LKH3. As shown in Figure 2, the probability of selecting the next city from the top 5 nearest neighbors exceeds 94%, increasing to over 99% for the top 10, and surpassing 99.9% for the top 15. Importantly, this distribution pattern remains consistent across different TSP sizes, suggesting a universal rule applicable to various scales.

Leveraging insights from the optimal solution, we construct the heatmap by assigning probabilities to edges based on the empirical distribution of the $k$-nearest prior $\hat{\mathbb{P}}_N(\cdot)$. For each city $i$ in a TSP instance of size $N$, we assign probabilities to edges $(i, j)$ as follows:

$$P_{ij}^N = \hat{\mathbb{P}}_N(k_{ij}), k_{ij} \in \{1, 2, ..., N\} \tag{6}$$

where $k_{ij}$ is the rank of city $j$ among $i$'s neighbors in terms of proximity (see the detailed statistical results in Appendix H). Importantly, this heatmap is parameter-free and scale-independent, thus requiring no tuning or learning phase.

## 5.2 PERFORMANCE DEMONSTRATION

**Experimental Setup** We conducted hyperparameter tuning for our proposed parameter-free heatmap, named GT-Prior, using the same setup and metrics as described in Section 4.3.

Table 3: Results on large-scale TSP problems. Abbreviations: RL (Reinforcement learning), SL (Supervised learning), UL (Unsupervised learning), AS (Active search), G (Greedy decoding), S (Sampling decoding), and BS (Beam-search). * indicates the baseline for performance gap calculation. † indicates methods utilizing heatmaps provided by Xia et al. (2024), with MCTS executed on our setup. Some methods list two terms for *Time*, corresponding to heatmap generation and MCTS runtimes, respectively. Baseline results (excluding those methods with MCTS) are sourced from Fu et al. (2021); Qiu et al. (2022).

| METHOD | TYPE | TSP-500 | | | TSP-1000 | | | TSP-10000 | | |
|---|---|---|---|---|---|---|---|---|---|---|
| | | LENGTH ↓ | GAP ↓ | TIME ↓ | LENGTH ↓ | GAP ↓ | TIME ↓ | LENGTH ↓ | GAP ↓ | TIME ↓ |
| CONCORDE | OR(EXACT) | 16.55* | — | 37.66M | 23.12* | — | 6.65H | N/A | N/A | N/A |
| GUROBI | OR(EXACT) | 16.55 | 0.00% | 45.63H | N/A | N/A | N/A | N/A | N/A | N/A |
| LKH-3 (DEFAULT) | OR | 16.55 | 0.00% | 46.28M | 23.12 | 0.00% | 2.57H | 71.78* | — | 8.8H |
| LKH-3 (LESS TRAILS) | OR | 16.55 | 0.00% | 3.03M | 23.12 | 0.00% | 7.73M | 71.79 | — | 51.27M |
| NEAREST INSERTION | OR | 20.62 | 24.59% | 0s | 28.96 | 25.26% | 0s | 90.51 | 26.11% | 6s |
| RANDOM INSERTION | OR | 18.57 | 12.21% | 0s | 26.12 | 12.98% | 0s | 81.85 | 14.04% | 4s |
| FARTHEST INSERTION | OR | 18.30 | 10.57% | 0s | 25.72 | 11.25% | 0s | 80.59 | 12.29% | 6s |
| EAN | RL+S | 28.63 | 73.03% | 20.18M | 50.30 | 117.59% | 37.07M | N/A | N/A | N/A |
| EAN | RL+S+2-OPT | 23.75 | 43.57% | 57.76M | 47.73 | 106.46% | 5.39H | N/A | N/A | N/A |
| AM | RL+S | 22.64 | 36.84% | 15.64M | 42.80 | 85.15% | 63.97M | 431.58 | 501.27% | 12.63M |
| AM | RL+G | 20.02 | 20.99% | 1.51M | 31.15 | 34.75% | 3.18M | 141.68 | 97.39% | 5.99M |
| AM | RL+BS | 19.53 | 18.03% | 21.99M | 29.90 | 29.23% | 1.64H | 129.40 | 80.28% | 1.81H |
| GCN | SL+G | 29.72 | 79.61% | 6.67M | 48.62 | 110.29% | 28.52M | N/A | N/A | N/A |
| GCN | SL+BS | 30.37 | 83.55% | 38.02M | 51.26 | 121.73% | 51.67M | N/A | N/A | N/A |
| POMO+EAS-EMB | RL+AS | 19.24 | 16.25% | 12.80H | N/A | N/A | N/A | N/A | N/A | N/A |
| POMO+EAS-LAY | RL+AS | 19.35 | 16.92% | 16.19H | N/A | N/A | N/A | N/A | N/A | N/A |
| POMO+EAS-TAB | RL+AS | 24.54 | 48.22% | 11.61H | 49.56 | 114.36% | 63.45H | N/A | N/A | N/A |
| ZERO | MCTS | 16.66 | 0.66% | **0.00M+1.67M** | 23.39 | 1.16% | **0.00M+3.34M** | 74.50 | 3.79% | **0.00M+16.78M** |
| ATT-GCN† | SL+MCTS | 16.66 | 0.69% | 0.52M+1.67M | 23.37 | 1.09% | 0.73M+3.34M | 73.95 | 3.02% | 4.16M+16.77M |
| DIMES† | RL+MCTS | 16.66 | 0.43% | 0.97M+1.67M | 23.37 | 1.11% | 2.08M+3.34M | 73.97 | 3.05% | 4.65M+16.77M |
| UTSP† | UL+MCTS | 16.69 | 0.90% | 1.37M+1.67M | 23.47 | 1.53% | 3.35M+3.34M | — | — | — |
| SOFTDIST† | SOFTDIST+MCTS | 16.62 | 0.43% | **0.00M+1.67M** | 23.30 | 0.80% | **0.00M+3.34M** | 73.89 | 2.94% | **0.00M+16.78M** |
| DIFUSCO† | SL+MCTS | **16.60** | **0.33%** | 3.61M+1.67M | **23.24** | **0.53%** | 11.86M+3.34M | 73.47 | 2.36% | 28.51M+16.87M |
| GT-PRIOR | PRIOR+MCTS | 16.63 | 0.50% | **0.00M+1.67M** | 23.31 | 0.85% | **0.00M+3.34M** | **73.31** | **2.13%** | **0.00M+16.78M** |

**Baselines** We evaluated several baseline methods in addition to those listed in Section 4.3. These include exact solvers such as Concorde (Applegate et al., 2009) and Gurobi (Gurobi Optimization, LLC, 2024) (using mixed-integer linear programming formulation), the heuristic solver LKH-3 (Helsgaun, 2017), and four end-to-end learning-based methods: EAN (d O Costa et al., 2020), AM (Kool et al., 2019), GCN (Joshi et al., 2019), and POMO+EAS (Hottung et al., 2021).

**Results** As detailed in Table 3, our simple heatmap construction method, combined with well-tuned MCTS, demonstrates competitive and often superior performance compared to more complex, learning-based approaches. The GT-Prior method exhibits remarkable consistency across different problem scales. For TSP-500, TSP-1000, and TSP-10000 instances, it consistently achieves solutions within 0.5%, 0.85%, and 2.13% of the best known solutions, respectively. Moreover, the GT-Prior method demonstrates a significant computational advantage. For instance, in TSP-10000 instances, our method achieved solutions within 2.13% of the best known, while reducing computational time by over 60% compared to the leading deep learning method DIFUSCO. This efficiency is partially due to our method not requiring heatmap generation, similar to the Zero and SoftDist baselines. This efficiency gain becomes increasingly important as problem sizes scale up, making our approach particularly suitable for large-scale TSP instances.

Interestingly, the Zero baseline, utilizing a heatmap filled entirely with zeros, provides further insights. Despite its apparent lack of guidance, it achieves surprisingly competitive results through careful hyperparameter tuning, particularly for TSP-500 and TSP-1000 instances. The most impactful hyperparameter for the Zero baseline is `Use_Heatmap`, with its optimal value being `False`, directing MCTS to construct candidate sets based on distance information rather than the uninformative heatmap. The strong performance of this baseline underscores the power of well-tuned search strategies, even without informative priors.

These results challenge the notion that sophisticated heatmap generation is necessary for effective TSP solving (Sun & Yang, 2023), aligning with the observation in SoftDist (Xia et al., 2024). It suggests that a judicious combination of a simple, statistically-informed heatmap with optimized

Table 4: Generalization performance of different methods trained on TSP500 across varying TSP sizes (TSP-500, TSP-1000, TSP-10000). "Res Type" refers to the result type: "Ori." indicates the performance on the same scales during the test phase, while "Gen." represents the model's generalized performance on different scales.

| METHOD | RES TYPE | TSP-500 | | TSP-1000 | | TSP-10000 | |
|--------|----------|---------|-------------|----------|-------------|-----------|-------------|
| | | GAP ↓ | DEGENERATION ↓ | GAP ↓ | DEGENERATION ↓ | GAP ↓ | DEGENERATION ↓ |
| DIMES | ORI. | 0.43% | 0.00% | 1.11% | 0.08% | 3.05% | 1.24% |
| | GEN. | 0.43% | | 1.19% | | 4.29% | |
| UTSP | ORI. | 0.90% | 0.00% | 1.53% | -0.09% | — | — |
| | GEN. | 0.90% | | 1.44% | | | |
| DIFUSCO | ORI. | 0.33% | 0.00% | 0.53% | 0.33% | 2.36% | 2.91% |
| | GEN. | 0.33% | | 0.86% | | 5.27% | |
| SOFTDIST | ORI. | 0.43% | 0.00% | 0.80% | 0.17% | 2.94% | 0.96% |
| | GEN. | 0.43% | | 0.97% | | 3.90% | |
| GT-PRIOR | ORI. | 0.50% | 0.00% | 0.85% | 0.03% | 2.13% | -0.01% |
| | GEN. | 0.50% | | 0.88% | | 2.13% | |

search strategies can yield highly competitive results, potentially shifting the focus in future research towards more balanced algorithm designs.

## 5.3 GENERALIZATION ABILITY

We evaluated the generalization ability of our parameter-free baseline, GT-Prior, against other methods across various TSP sizes. The MCTS of each method employs the corresponding Tuned setting as described in Section 4.3. Table 4 presents the generalization performance of models trained on TSP-500. GT-Prior demonstrates superior generalization performance across all problem scales. For TSP-1000, GT-Prior exhibits minimal performance degradation (0.01%) relative to other methods. Remarkably, for TSP-10000, GT-Prior maintains consistent performance with a slight improvement (-0.01% degradation), surpassing all other approaches. Conversely, DIMES, DIFUSCO, and SoftDist exhibit increasing performance degradation as problem size increases, with DIFUSCO experiencing the most substantial decline (2.91%) for TSP-10000. These results highlight the robust generalization capability of GT-Prior, especially for larger problem instances. The generalization results of models trained on TSP-1000 and TSP-10000 are left in Appendix D, and additional results on TSPLIB instances are listed in Appendix E.

## 6 ABLATION STUDY

To better understand the efficacy of hyperparameter tuning in MCTS for solving TSP, we conducted an ablation study focusing on two critical aspects: the relationship between search time and solution quality, and the sample efficiency of our tuning process. These experiments provide valuable insights into our algorithm's performance characteristics and highlight areas for potential optimization.

### 6.1 IMPACT OF TUNING STAGE TIME_LIMIT ON SOLVER PERFORMANCE

The relationship between Time_Limit and hyperparameter quality is crucial in MCTS hyperparameter tuning. While longer search times might intuitively yield better results, they also lead to significantly increased tuning time. We conducted an ablation study to investigate this trade-off and seek a balance between performance and efficiency.

**Experimental Setup** We examined the impact of search time on solver performance for TSP-500 and TSP-1000 instances, varying the tuning stage Time_Limit from 0.1 to 0.05 and 0.01.

Figure 3 shows the performance of different methods with varying inference times, each with three hyperparameter sets tuned using different Time_Limit values. Surprisingly, the relative performance remains largely consistent across search durations, suggesting that hyperparameter effectiveness can be accurately assessed within a limited time frame.

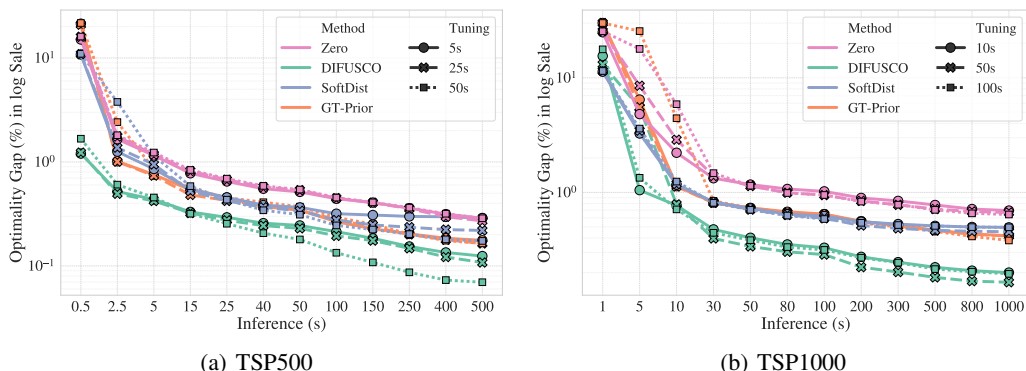

Figure 3: Impact of search time on solver performance across different hyperparameter configurations

For TSP-500, most heatmaps exhibit similar performance across all tuning stage `Time_Limit` values, with Zero and GT-Prior methods showing nearly identical performance curves. The best learning-based method, DIFUSCO, displays a small performance gap at the default 50-second inference time limit. However, this gap widens with longer inference times, suggesting that optimal MCTS settings for high-quality heatmaps may vary with different `Time_Limit` values during tuning phase. Efficiently tuning hyperparameters for such high-quality heatmaps remains a future research direction. Notably, TSP-1000 results show even smaller performance gaps between different tuning stage `Time_Limit` values, indicating that shorter tuning times can yield satisfactory hyperparameter settings for larger problem instances.

The consistency of relative performance across search times has significant implications for efficient hyperparameter tuning in large-scale TSP solving. This insight enables the development of accelerated evaluation procedures that can identify promising hyperparameter settings without exhaustive, long-duration searches.

## 6.2 SAMPLE EFFICIENCY

Experiments were conducted to evaluate the sample efficiency of the hyperparameter tuning procedure for our proposed $k$-nearest prior heatmap. By varying the number of TSP instances in the training set and measuring the resulting solution quality of the tuned hyperparameter setting, insights were gained into the computational efficiency of our method. With only 64 samples for hyperparameter tuning, our proposed GT-prior achieved a gap of 0.493% on TSP-500 and 0.866% on TSP-1000, rivaling the performance of hyperparameter tuning with 256 samples, which achieved 0.493% on TSP-500 and 0.858% on TSP-1000. These results demonstrate the high sample efficiency of our approach, enabling effective tuning with minimal computational resources.

## 7 CONCLUSIONS AND LIMITATIONS

This study revisited the "Heatmap + MCTS" paradigm for large-scale TSP, highlighting the under-estimated importance of MCTS hyperparameter tuning. We demonstrated that careful tuning, especially of parameters like `Max_Candidate_Num`, can drastically improve solution quality, even with simple or non-informative heatmaps. To this end, we introduced a parameter-free $k$-nearest prior heatmap, which achieves competitive performance against complex learning-based methods across various TSP sizes. This simple yet effective approach challenges the prevailing focus on sophisticated models, showing that leveraging basic statistical prior of TSP can often be sufficient, particularly when scaling to large instances. Future work should explore more adaptive search strategies within MCTS or improve tuning efficiency through advanced optimization techniques.

Overall, this study contributes a nuanced understanding that could pivot future research towards more balanced and efficacious integration of learning and search in TSP algorithms.

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

## A  ADDITIONAL RELATED WORKS

Approaches using machine learning to address the Travelling Salesman Problem (TSP) generally fall into two distinct groups based on how they generate solutions. The first group, known as construction methods, incrementally forms a path by sequentially adding cities to an unfinished route, following an autoregressive process until the entire path is completed. The second group, improvement methods, starts with a complete route and continually applies local search operations to improve the solution.

**Construction Methods**   Since Vinyals et al. (2015); Bello et al. (2016) introduced the autoregressive combinatorial optimization neural solver, numerous advancements have emerged in subsequent years (Deudon et al., 2018; Kool et al., 2019; Peng et al., 2020; Kwon et al., 2021; 2020). These include enhanced network architectures (Kool et al., 2019), more sophisticated deep reinforcement learning techniques (Khalil et al., 2017; Ma et al., 2019; Choo et al., 2022), and improved training methods (Kim et al., 2022; Bi et al., 2022). For large-scale TSP, Pan et al. (2023) adopts a hierarchical divide-and-conquer approach, breaking down the complex TSP into more manageable open-loop TSP sub-problems.

**Improvement Methods**   In contrast to construction methods, improvement-based solvers leverage neural networks to progressively refine an existing feasible solution, continuing the process until the computational limit is reached. These improvement methods are often influenced by traditional local search techniques like $k$-opt, and have been shown to deliver impressive results in various previous studies (Chen & Tian, 2019; Wu et al., 2021; Kim et al., 2021; Hudson et al., 2021). Ye et al. (2024) implements a divide-and-conquer approach, using search-based methods to enhance the solutions of smaller subproblems generated from the larger instances.

Recent breakthroughs in solving large-scale TSP problems (Fu et al., 2021; Qiu et al., 2022; Sun & Yang, 2023; Min et al., 2024; Xia et al., 2024), have incorporated Monte Carlo tree search (MCTS) as an effective post-processing technique. These heatmaps serve as priors for guiding the MCTS, resulting in impressive performance in large-scale TSP solutions, achieving state-of-the-art results.

**Other Directions**   In addition to exploring solution methods for combinatorial optimization problems, some studies investigate intrinsic challenges encountered during the learning phase. These include generalization issues during inference (Wang et al., 2021; Zhou et al., 2023; Wang et al., 2024) and multi-task learning (Wang & Yu, 2023; Liu et al., 2024; Zhou et al., 2024) aimed at developing foundational models.

## B  IMPACT OF HEURISTIC POSTPROCESSING

In our experimental reproduction of various learning-based heatmap generation methods for the Travelling Salesman Problem (TSP), we identified a critical yet often overlooked factor affecting performance: the post-processing of model-generated heatmaps. This section details the post-processing strategies employed by different methods and evaluates their impact on performance metrics.

### B.1  POSTPROCESSING STRATEGIES

**DIMES**   DIMES generates an initial sparse heatmap matrix of dimension $n \times n$ from a $k$-nearest neighbors ($k$-NN) subgraph of the original TSP instance ($k = 50$). The post-processing involves two steps:

1. Sparsification: Retaining only the top-5 values for each row, setting all others to a significantly negative number.

2. Adaptive $\mathrm{softmax}$: Iteratively applying a temperature-scaled $\mathrm{softmax}$ function with gradual temperature reduction until the minimum non-zero probability exceeds a predefined threshold.

**DIFUSCO**   DIFUSCO also generates a sparse heatmap based on the $k$-NN subgraph ($k = 50$ for TSP-500, $k = 100$ for larger scales). The post-processing differs based on problem scale:

Table 5: Performance Degeneration for Different Methods with and without Postprocessing on TSP-500, TSP-1000, and TSP-10000. 'W' indicates with postprocessing, while 'W/O' indicates without postprocessing.

| METHOD | POSTPROCESSING | TSP-500 | | TSP-1000 | | TSP-10000 | |
|---|---|---|---|---|---|---|---|
| | | GAP ↓ | DEGENERATIONS ↓ | GAP ↓ | DEGENERATIONS ↓ | GAP ↓ | DEGENERATIONS ↓ |
| DIMES | W/O
W | 2.50%
1.57% | 0.93% | 9.07%
2.30% | 6.77% | 15.87%
3.05% | 12.81% |
| UTSP | W/O
W | 4.50%
3.14% | 1.36% | 6.30%
4.20% | 2.10% | —
 | — |
| DIFUSCO | W/O
W | 2.33%
0.45% | 1.88% | 0.66%
1.07% | -0.40% | 45.20%
2.69% | 42.52% |

1. For TSP-500 and TSP-1000: A single step integrating Euclidean distances, thresholding, and symmetrization.

2. For TSP-10000: Two steps are applied sequentially: a) Additional supervision using a greedy decoding strategy followed by 2-opt heuristics. b) The same process as used for smaller instances.

**UTSP** UTSP's post-processing is straightforward, involving sparsification of the dense heatmap matrix by preserving only the top 20 values per row.

## B.2 EXPERIMENTAL RESULTS

We conducted experiments on the test set for heatmaps generated by these three methods, both with and without post-processing, using the default MCTS setting. Results are presented in Table 5.

Our findings reveal that heatmaps generated without post-processing generally exhibit performance degradation, particularly for TSP-10000, where the gap increases by orders of magnitude. This underscores the importance of sparsification for large-scale instances and highlights the tendency of existing methodologies to overstate their efficacy in training complex deep learning models.

Interestingly, DIFUSCO's heatmap without post-processing outperforms its post-processed counterpart for TSP-1000, suggesting that the DIFUSCO model, when well-trained on this scale, can generate helpful heatmap matrices to guide MCTS without additional refinement.

These results emphasize the critical role of post-processing in enhancing the performance of learning-based heatmap generation methods for TSP, particularly as problem scales increase. They also highlight the need for careful evaluation of model outputs and the potential for over-reliance on post-processing to mask limitations in model training and generalization.

The substantial performance gap between heatmaps with and without post-processing raises questions about the extent to which the reported performance gains can be attributed solely to the learning modules of these methods. While the learning components undoubtedly contribute to the overall effectiveness, the significant impact of post-processing suggests that the raw output of the learning models may not be as refined or directly applicable as previously thought.

In light of these findings, we recommend that future research on heatmap-based methods for TSP provide a detailed description of their post-processing operations. Additionally, we suggest reporting results both with and without post-processing to offer a more comprehensive understanding of the method's performance and the relative contributions of its learning and post-processing components. This approach would foster greater transparency in the field and facilitate more accurate comparisons between different methodologies.

## C ADDITIONAL HYPERPARAMETER IMPORTANCE ANALYSIS

The SHAP method was employed to provide more insights into hyperparameter importance for all conducted grid search experiments. Most of the beeswarm plots for TSP-500, TSP-1000, and

TSP-10000 are presented in Figures 4. The beeswarm plots for the Zero heatmap are presented in Figure 5, as only the case where `Use_Heatmap` is set to `False` is considered for the Zero heatmap.

The patterns of TSP-1000 are similar to those of TSP-500, as discussed in Section 4.2. However, the patterns for TSP-10000 show a major difference, where the influence of `Max_Candidate_Num` and `Use_Heatmap` becomes dominant. Furthermore, their SHAP values are clearly clustered rather than continuous, as observed in smaller scales. This could be explained by the candidate set of large-scale TSP instances having a major impact on the running time of MCTS $k$-opt search. Additionally, the time limit setting causes the performance of different hyperparameter settings for `Max_Candidate_Num` and `Use_Heatmap` to become more distinct.

## D  ADDITIONAL GENERALIZATION ABILITY RESULTS

Tables 6 presents additional results on the generalization ability of various methods when trained on TSP-1000 and TSP-10000, respectively.

For models trained on TSP-1000, GT-Prior continues to demonstrate superior generalization capability. When generalizing to smaller instances (TSP-500), GT-Prior shows minimal performance degradation (0.02%), comparable to DIMES and better than UTSP and SoftDist. For larger instances (TSP-10000), GT-Prior maintains consistent performance with a slight improvement (-0.02% degradation), outperforming all other methods. DIFUSCO, while showing good performance on TSP-500 and TSP-1000, experiences significant degradation (2.91%) when scaling to TSP-10000.

The results for models trained on TSP-10000 further highlight GT-Prior's robust generalization ability. When applied to smaller problem sizes (TSP-500 and TSP-1000), GT-Prior exhibits minimal performance degradation (0.01% and 0.02%, respectively). In contrast, other methods show more substantial degradation, particularly for TSP-1000. Notably, SoftDist experiences severe performance deterioration (73.36%) when generalizing to TSP-1000, while DIFUSCO shows significant degradation for both TSP-500 (0.63%) and TSP-1000 (2.74%).

These results consistently demonstrate GT-Prior's exceptional ability to generalize across various problem scales, maintaining stable performance regardless of whether it is scaling up or down from the training instance size. This stability is particularly evident when compared to the other methods, which often struggle with significant performance degradation when generalizing to different problem sizes.

## E  ADDITIONAL RESULTS ON TSPLIB

We categorize all Euclidean 2D TSP instances into three groups based on the number of nodes: Small (0-500 nodes), Medium (500-2000 nodes), and Large (more than 2000 nodes). For each category, we evaluate all baseline methods alongside our proposed GT-Prior.

We conducted MCTS evaluations under two distinct parameter settings: (1) Tuned Settings, optimized using uniform TSP instances as listed in Table 14, whose results are shown in Table 7, 8, 9, and (2) the Default Settings, as originally employed by Fu et al. (2021), whose results are shown in Table 10, 11, 12. The results in these tables showcase the performance of the methods in terms of solution length and optimality gap, highlighting the effectiveness of the proposed GT-Prior approach.

Several key insights emerge from these experimental results. First, we observe a strong interaction between instance distribution and parameter tuning effectiveness. While methods like UTSP and DIMES excel on small uniform instances, their performance exhibits high sensitivity to parameter settings when faced with real-world TSPLIB instances, particularly at larger scales (e.g., UTSP degrading from 26.51% to 1481.66% on large instances). This finding reveals a fundamental generalization challenge shared by most learning-based methods - the optimal parameters learned from one distribution may not transfer effectively to another, highlighting the critical importance of robust parameter tuning strategies. To illustrate this distribution sensitivity, we visualize representative hard and easy instances from each group in Figures 6, demonstrating that hard instances deviate significantly from uniform distribution while easy instances closely resemble it.

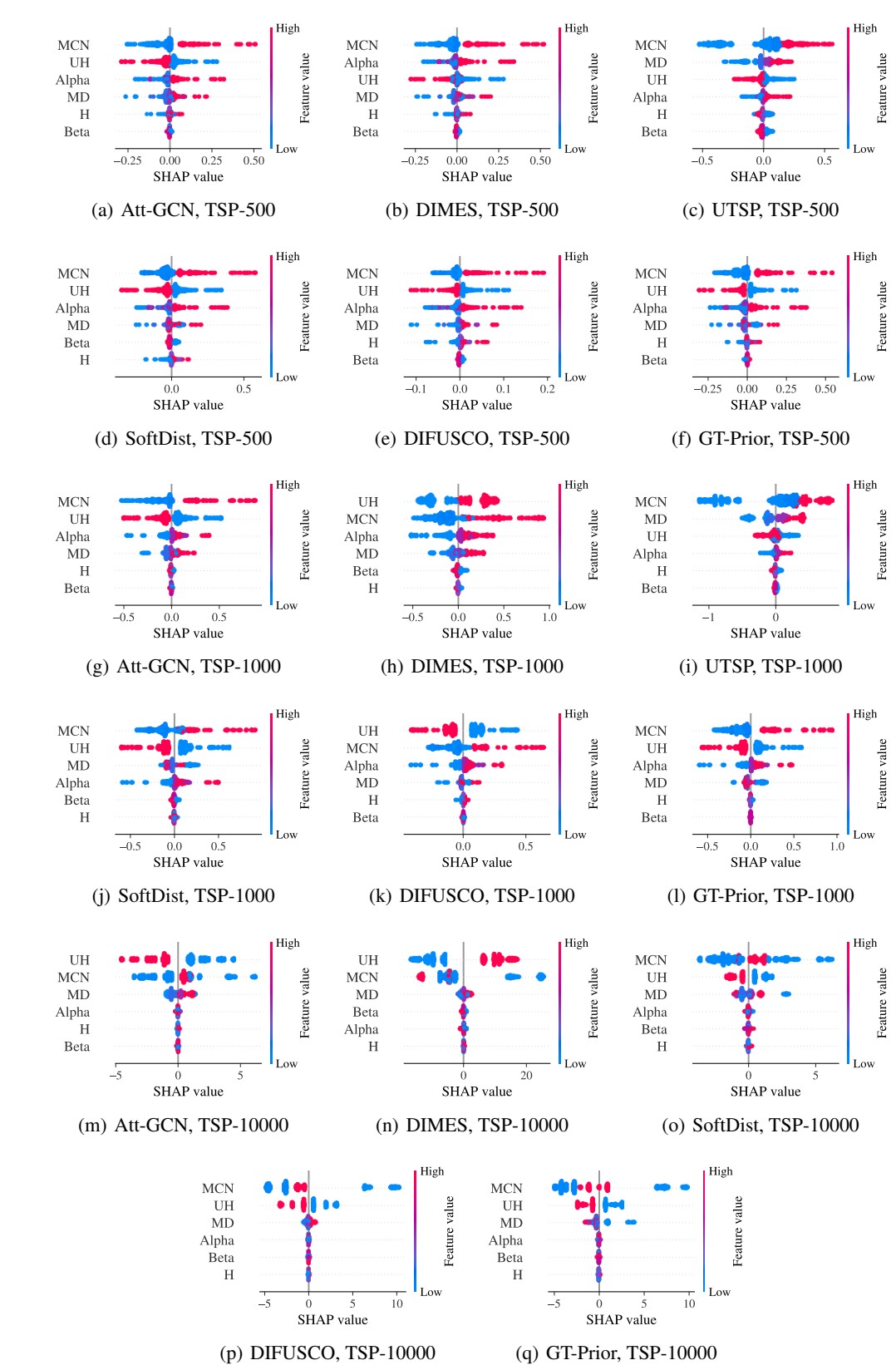

Figure 4: Beeswarm plots of SHAP values for Att-GCN, DIMES, SoftDist, DIFUSCO and GT-Prior.

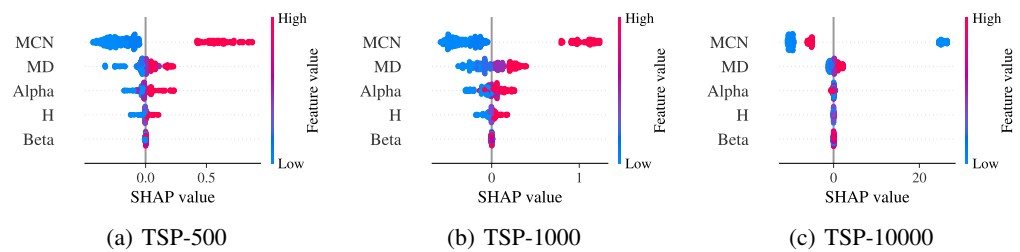

Figure 5: Beeswarm plots of SHAP values for Zero Heatmap.

Table 6: Generalization on the model trained on TSP1000 (the upper table) and TSP10000 (the lower table).

| METHOD | RES TYPE | TSP-500 | | TSP-1000 | | TSP-10000 | |
|---|---|---|---|---|---|---|---|
| | | GAP ↓ | DEGENERATION ↓ | GAP ↓ | DEGENERATION ↓ | GAP ↓ | DEGENERATION ↓ |
| DIMES | ORI. | 0.69% | 0.02% | 1.11% | 0.00% | 3.05% | 1.01% |
| | GEN. | 0.71% | | 1.11% | | 4.06% | |
| UTSP | ORI. | 0.90% | 0.06% | 1.53% | 0.00% | — | — |
| | GEN. | 0.96% | | 1.53% | | | |
| DIFUSCO | ORI. | 0.33% | -0.07% | 0.53% | 0.00% | 2.36% | 2.91% |
| | GEN. | 0.26% | | 0.53% | | 5.27% | |
| SOFTDIST | ORI. | 0.43% | 0.08% | 0.80% | 0.00% | 2.94% | 0.74% |
| | GEN. | 0.51% | | 0.80% | | 3.68% | |
| GT-PRIOR | ORI. | 0.50% | 0.02% | 0.85% | 0.00% | 2.13% | -0.02% |
| | GEN. | 0.52% | | 0.85% | | 2.11% | |

| METHOD | RES TYPE | TSP-500 | | TSP-1000 | | TSP-10000 | |
|---|---|---|---|---|---|---|---|
| | | GAP ↓ | DEGENERATION ↓ | GAP ↓ | DEGENERATION ↓ | GAP ↓ | DEGENERATION ↓ |
| DIMES | ORI. | 0.69% | 0.06% | 1.11% | 0.07% | 3.05% | 0.00% |
| | GEN. | 0.75% | | 1.18% | | 3.05% | |
| DIFUSCO | ORI. | 0.33% | 0.63% | 0.53% | 2.81% | 2.36% | 0.00% |
| | GEN. | 0.95% | | 3.34% | | 2.36% | |
| SOFTDIST | ORI. | 0.43% | 0.22% | 0.80% | 73.44% | 2.94% | 0.00% |
| | GEN. | 0.65% | | 74.24% | | 2.94% | |
| GT-PRIOR | ORI. | 0.50% | 0.01% | 0.85% | 0.04% | 2.13% | 0.00% |
| | GEN. | 0.51% | | 0.89% | | 2.13% | |

This generalization issue is particularly noteworthy as it affects all methods except the Zero heatmap, which maintains relatively stable performance across different instance sizes and parameter settings. The Zero heatmap's consistency (varying only from 5.54% to 6.51% on large instances) provides compelling evidence for our thesis that the MCTS component's contribution to solution quality has been historically undervalued in the framework. Furthermore, this stability suggests that proper MCTS parameter tuning might be more crucial for achieving robust performance than developing increasingly sophisticated heatmap generation methods.

From a practical perspective, our analysis also reveals an important computational consideration. The learning-based baselines necessitate GPU resources for both training and inference stages, potentially creating a bottleneck when dealing with real-world data. In contrast, methods that reduce reliance on complex learned components might offer more practical utility in resource-constrained settings while maintaining competitive performance through careful parameter optimization.

These findings collectively suggest that future research in this domain might benefit from a more balanced focus between heatmap sophistication and MCTS optimization, particularly when considering real-world applications where robustness and computational efficiency are paramount.

Table 7: Results on small TSPLIB instances (with 0-500 nodes). The hyperparameter settings are tuned on uniform TSP instances as listed in Table 14.

| Instance | Optimal | Zero Length↓ | Gap↓ | Att-GCN Length↓ | Gap↓ | DIMES Length↓ | Gap↓ | UTSP Length↓ | Gap↓ | SoftDist Length↓ | Gap↓ | DIFUSCO Length↓ | Gap↓ | GT-Prior Length↓ | Gap↓ |
|---|---|---|---|---|---|---|---|---|---|---|---|---|---|---|---|
| st70 | 675 | **676** | **0.15%** | **676** | **0.15%** | **676** | **0.15%** | **676** | **0.15%** | 724 | 7.26% | **676** | **0.15%** | **676** | **0.15%** |
| eil76 | 538 | **538** | **0.00%** | **538** | **0.00%** | **538** | **0.00%** | **538** | **0.00%** | **538** | **0.00%** | **538** | **0.00%** | **538** | **0.00%** |
| kroA200 | 29368 | **29368** | **0.00%** | 29383 | 0.05% | **29368** | **0.00%** | 29382 | 0.05% | 29383 | 0.05% | 29380 | 0.04% | **29368** | **0.00%** |
| eil51 | 426 | **427** | **0.23%** | **427** | **0.23%** | **427** | **0.23%** | **427** | **0.23%** | **427** | **0.23%** | **427** | **0.23%** | **427** | **0.23%** |
| rat195 | 2323 | 2328 | 0.22% | 2328 | 0.22% | **2323** | **0.00%** | 2328 | 0.22% | 2328 | 0.22% | 2328 | 0.22% | 2328 | 0.22% |
| pr144 | 58537 | 59932 | 2.38% | 63736 | 8.88% | 59553 | 1.74% | **59211** | **1.15%** | 66950 | 14.37% | 63389 | 8.29% | 65486 | 11.87% |
| bier127 | 118282 | **118282** | **0.00%** | **118282** | **0.00%** | **118282** | **0.00%** | **118282** | **0.00%** | **118282** | **0.00%** | **118282** | **0.00%** | **118282** | **0.00%** |
| lin105 | 14379 | **14379** | **0.00%** | **14379** | **0.00%** | **14379** | **0.00%** | **14379** | **0.00%** | 15081 | 4.88% | **14379** | **0.00%** | **14379** | **0.00%** |
| kroD100 | 21294 | **21294** | **0.00%** | **21294** | **0.00%** | **21294** | **0.00%** | **21294** | **0.00%** | **21294** | **0.00%** | **21294** | **0.00%** | **21294** | **0.00%** |
| kroA100 | 21282 | **21282** | **0.00%** | **21282** | **0.00%** | **21282** | **0.00%** | **21282** | **0.00%** | **21282** | **0.00%** | **21282** | **0.00%** | **21282** | **0.00%** |
| pr152 | 73682 | 74089 | 0.55% | **73682** | **0.00%** | **73682** | **0.00%** | 73818 | 0.18% | 74443 | 1.03% | 74609 | 1.26% | 74274 | 0.80% |
| ts225 | 126643 | **126643** | **0.00%** | **126643** | **0.00%** | **126643** | **0.00%** | **126643** | **0.00%** | **126643** | **0.00%** | **126643** | **0.00%** | **126643** | **0.00%** |
| rd400 | 15281 | 15314 | 0.22% | 15333 | 0.34% | 15323 | 0.27% | 15408 | 0.83% | 15352 | 0.46% | 15320 | 0.26% | **15303** | **0.14%** |
| kroB100 | 22141 | **22141** | **0.00%** | **22141** | **0.00%** | **22141** | **0.00%** | **22141** | **0.00%** | **22141** | **0.00%** | **22141** | **0.00%** | **22141** | **0.00%** |
| d198 | 15780 | 15817 | 0.23% | 16344 | 3.57% | 15844 | 0.41% | **15804** | **0.15%** | 15816 | 0.23% | 16237 | 2.90% | 15817 | 0.23% |
| eil101 | 629 | **629** | **0.00%** | **629** | **0.00%** | **629** | **0.00%** | **629** | **0.00%** | **629** | **0.00%** | **629** | **0.00%** | **629** | **0.00%** |
| linhp318 | 41345 | 42558 | 2.93% | 42523 | 2.85% | 42763 | 3.43% | 42420 | 2.60% | 42283 | 2.27% | **42223** | **2.12%** | 42387 | 2.52% |
| gil262 | 2378 | 2380 | 0.08% | 2382 | 0.17% | 2380 | 0.08% | 2380 | 0.08% | **2379** | **0.04%** | 2380 | 0.08% | 2380 | 0.08% |
| rat99 | 1211 | **1211** | **0.00%** | **1211** | **0.00%** | **1211** | **0.00%** | **1211** | **0.00%** | **1211** | **0.00%** | **1211** | **0.00%** | **1211** | **0.00%** |
| berlin52 | 7542 | **7542** | **0.00%** | **7542** | **0.00%** | **7542** | **0.00%** | **7542** | **0.00%** | **7542** | **0.00%** | **7542** | **0.00%** | **7542** | **0.00%** |
| kroC100 | 20749 | **20749** | **0.00%** | **20749** | **0.00%** | **20749** | **0.00%** | **20749** | **0.00%** | **20749** | **0.00%** | **20749** | **0.00%** | **20749** | **0.00%** |
| pr226 | 80369 | 87311 | 8.64% | 83828 | 4.30% | 83828 | 4.30% | 81058 | 0.86% | 80850 | 0.60% | **80463** | **0.12%** | 85793 | 6.75% |
| fl417 | 11861 | 12852 | 8.36% | **11945** | **0.71%** | 12169 | 2.60% | 12800 | 7.92% | 13198 | 11.27% | 12158 | 2.50% | 12437 | 4.86% |
| kroE100 | 22068 | **22068** | **0.00%** | **22068** | **0.00%** | **22068** | **0.00%** | **22068** | **0.00%** | **22068** | **0.00%** | **22068** | **0.00%** | **22068** | **0.00%** |
| pr76 | 108159 | **108159** | **0.00%** | **108159** | **0.00%** | **108159** | **0.00%** | **108159** | **0.00%** | 109325 | 1.08% | **108159** | **0.00%** | **108159** | **0.00%** |
| ch130 | 6110 | **6111** | **0.02%** | **6111** | **0.02%** | **6111** | **0.02%** | **6111** | **0.02%** | 6242 | 2.16% | **6111** | **0.02%** | **6111** | **0.02%** |
| tsp225 | 3916 | 3932 | 0.41% | **3916** | **0.00%** | 3919 | 0.08% | 3923 | 0.18% | **3916** | **0.00%** | **3916** | **0.00%** | 3923 | 0.18% |
| rd100 | 7910 | **7910** | **0.00%** | **7910** | **0.00%** | **7910** | **0.00%** | **7910** | **0.00%** | 7938 | 0.35% | **7910** | **0.00%** | **7910** | **0.00%** |
| pr264 | 49135 | 51267 | 4.34% | 50451 | 2.68% | 49949 | 1.66% | 49635 | 1.02% | **49374** | **0.49%** | 50389 | 2.55% | 49508 | 0.76% |
| pr124 | 59030 | 59168 | 0.23% | 59210 | 0.30% | 59551 | 0.88% | 59210 | 0.30% | 59257 | 0.38% | 59688 | 1.11% | **59030** | **0.00%** |
| kroA150 | 26524 | **26525** | **0.00%** | **26525** | **0.00%** | **26525** | **0.00%** | **26525** | **0.00%** | **26525** | **0.00%** | **26525** | **0.00%** | **26525** | **0.00%** |
| kroB200 | 29437 | **29437** | **0.00%** | 29438 | 0.00% | **29437** | **0.00%** | 29446 | 0.03% | **29437** | **0.00%** | **29437** | **0.00%** | **29437** | **0.00%** |
| kroB150 | 26130 | 26178 | 0.18% | 26141 | 0.04% | 26176 | 0.18% | 26131 | 0.02% | **26130** | **0.00%** | 26143 | 0.05% | **26130** | **0.00%** |
| pr107 | 44303 | **44303** | **0.00%** | 44387 | 0.19% | **44303** | **0.00%** | **44303** | **0.00%** | **44303** | **0.00%** | **44303** | **0.00%** | **44303** | **0.00%** |
| lin318 | 42029 | 42558 | 1.26% | 42561 | 1.27% | 42609 | 1.38% | 42420 | 0.93% | 42283 | 0.60% | **42254** | **0.54%** | 42387 | 0.85% |
| pr136 | 96772 | **96772** | **0.00%** | **96772** | **0.00%** | **96772** | **0.00%** | **96772** | **0.00%** | **96772** | **0.00%** | **96772** | **0.00%** | **96772** | **0.00%** |
| pr299 | 48191 | 48279 | 0.18% | 48223 | 0.07% | 48230 | 0.08% | **48191** | **0.00%** | 48197 | 0.01% | 48269 | 0.16% | 48197 | 0.01% |
| u159 | 42080 | **42080** | **0.00%** | **42080** | **0.00%** | **42080** | **0.00%** | **42080** | **0.00%** | 42396 | 0.75% | **42080** | **0.00%** | **42080** | **0.00%** |
| a280 | 2579 | **2579** | **0.00%** | **2579** | **0.00%** | **2579** | **0.00%** | **2579** | **0.00%** | **2579** | **0.00%** | **2579** | **0.00%** | **2579** | **0.00%** |
| pr439 | 107217 | 109241 | 1.89% | 108944 | 1.61% | 109624 | 2.22% | **108476** | **1.17%** | 110701 | 3.25% | 108485 | 1.18% | 109624 | 2.24% |
| ch150 | 6528 | **6528** | **0.00%** | **6528** | **0.00%** | **6528** | **0.00%** | **6528** | **0.00%** | 6533 | 0.08% | **6528** | **0.00%** | **6528** | **0.00%** |
| d493 | 35002 | 35347 | 0.99% | 35331 | 0.94% | 35318 | 0.90% | **35235** | **0.67%** | 35297 | 0.84% | 35292 | 0.83% | 35244 | 0.69% |
| pcb442 | 50778 | 50935 | 0.31% | 50902 | 0.24% | 50856 | 0.15% | 51060 | 0.56% | **50847** | **0.14%** | 50908 | 0.26% | 50927 | 0.29% |
| Average | - | 35281 | 0.79% | 35244 | 0.67% | 35155 | 0.48% | **35050** | **0.45%** | 35340 | 1.23% | 35165 | 0.58% | 35321 | 0.76% |

Table 8: Results on medium TSPLIB instances (with 500-2000 nodes). The hyperparameter settings are tuned on uniform TSP instances as listed in Table 14.

| Instance | Optimal | Zero Length↓ | Gap↓ | Att-GCN Length↓ | Gap↓ | DIMES Length↓ | Gap↓ | UTSP Length↓ | Gap↓ | SoftDist Length↓ | Gap↓ | DIFUSCO Length↓ | Gap↓ | GT-Prior Length↓ | Gap↓ |
|---|---|---|---|---|---|---|---|---|---|---|---|---|---|---|---|
| u574 | 36905 | 37211 | 0.83% | 37226 | 0.87% | 37399 | 1.34% | 37211 | 0.83% | 37142 | 0.64% | **36989** | **0.23%** | 37146 | 0.65% |
| pcb1173 | 56892 | 57837 | 1.66% | 57715 | 1.45% | 57618 | 1.28% | 57770 | 1.54% | 57633 | 1.30% | 57304 | 0.72% | **57248** | **0.63%** |
| rat783 | 8806 | 8903 | 1.10% | 8887 | 0.92% | 8892 | 0.98% | 8919 | 1.28% | 8884 | 0.89% | **8842** | **0.41%** | 8851 | 0.51% |
| u1432 | 152970 | 156669 | 2.42% | 154684 | 1.12% | 154889 | 1.25% | 154703 | 1.13% | 154338 | 0.89% | **154046** | **0.70%** | 154285 | 0.86% |
| fl1400 | 20127 | 27446 | 36.36% | 26280 | 30.57% | 23066 | 14.60% | 23467 | 16.59% | 29343 | 45.79% | **21519** | **6.92%** | 22924 | 13.90% |
| vm1084 | 239297 | 255009 | 6.57% | 257899 | 7.77% | 254512 | 6.36% | 246531 | 3.02% | **240016** | **0.30%** | 240265 | 0.40% | 244968 | 2.37% |
| rat575 | 6773 | 6844 | 1.05% | 6826 | 0.78% | 6845 | 1.06% | 6829 | 0.83% | 6814 | 0.61% | **6800** | **0.40%** | 6807 | 0.50% |
| vm1748 | 336556 | 377814 | 12.26% | 385587 | 14.57% | 378032 | 11.90% | 376605 | 11.90% | 341506 | 1.47% | **341443** | **1.45%** | 343834 | 2.16% |
| rl1889 | 316536 | 479282 | 51.41% | 444184 | 40.33% | 397609 | 25.61% | 441143 | 39.37% | 327774 | 3.55% | **324242** | **2.43%** | 451948 | 42.78% |
| u724 | 41910 | 42288 | 0.90% | 42105 | 0.47% | 42330 | 1.00% | 42317 | 0.97% | 42161 | 0.60% | **42003** | **0.22%** | 42086 | 0.42% |
| d1291 | 50801 | 72786 | 43.28% | 70051 | 37.89% | 71972 | 41.67% | 72779 | 43.26% | 52023 | 2.41% | **51342** | **1.06%** | 74911 | 47.46% |
| pr1002 | 259045 | 265784 | 2.60% | 265338 | 2.43% | 263164 | 1.59% | 264061 | 1.94% | 262591 | 1.37% | **262472** | **1.32%** | 262929 | 1.50% |
| fl1577 | 22249 | 29723 | 33.59% | 27605 | 24.07% | 30050 | 35.06% | 29581 | 32.95% | 29102 | 30.80% | **25960** | **16.68%** | 29222 | 31.34% |
| nrw1379 | 56638 | 57171 | 0.94% | 57070 | 0.76% | 57326 | 1.21% | 57172 | 0.94% | 58266 | 2.87% | **56961** | **0.57%** | 56974 | 0.59% |
| rl1304 | 252948 | 332691 | 31.53% | 316879 | 25.27% | 316925 | 25.29% | 316283 | 25.04% | 262598 | 3.82% | **257797** | **1.92%** | 297448 | 17.59% |
| d657 | 48912 | 49228 | 0.65% | 49228 | 0.65% | 49303 | 0.80% | 49350 | 0.90% | **49094** | **0.37%** | 49098 | 0.38% | 49118 | 0.42% |
| p654 | 34643 | 38112 | 10.01% | 38864 | 12.18% | **35210** | **1.64%** | 35884 | 3.58% | 47033 | 35.76% | 36765 | 6.13% | 35569 | 2.67% |
| d1655 | 62128 | 66466 | 6.98% | 65547 | 5.50% | 64743 | 4.21% | 65977 | 6.20% | 63986 | 2.99% | 64358 | 3.59% | **63951** | **2.93%** |
| u1817 | 57201 | 90599 | 58.39% | 68245 | 19.31% | 71276 | 24.61% | 80609 | 40.92% | 58838 | 2.86% | **58587** | **2.42%** | 75131 | 31.35% |
| u1060 | 224094 | 233417 | 4.16% | 232573 | 3.78% | 242781 | 8.34% | 236860 | 5.70% | 227830 | 1.67% | **225164** | **0.48%** | 229725 | 2.51% |
| rl1323 | 270199 | 306164 | 13.31% | 297453 | 10.09% | 305970 | 13.24% | 307474 | 13.80% | 274440 | 1.57% | **274104** | **1.45%** | 293294 | 8.55% |
| Average | - | 142449 | 15.24% | 138583 | 11.47% | 136662 | 10.64% | 138644 | 12.03% | 125305 | 6.79% | **123621** | **2.38%** | 135160 | 10.08% |

## F   TUNED HYPERPARAMETER SETTINGS

In this section, we present the results of hyperparameter tuning, summarized in the following Table 14. The table includes the various hyperparameter combinations explored during the tuning process and their corresponding heatmap generation methods.

Table 9: Results on large TSPLIB instances (with more than 2000 nodes). The hyperparameter settings are tuned on uniform TSP instances as listed in Table 14.

| Instance | Optimal | Zero | | Att-GCN | | DIMES | | UTSP | | SoftDist | | DIFUSCO | | GT-Prior | |
|---|---|---|---|---|---|---|---|---|---|---|---|---|---|---|---|
| | | Length↓ | Gap↓ | Length↓ | Gap↓ | Length↓ | Gap↓ | Length↓ | Gap↓ | Length↓ | Gap↓ | Length↓ | Gap↓ | Length↓ | Gap↓ |
| u2152 | 64253 | 66719 | 3.84% | 66301 | 3.19% | 67244 | 4.66% | 79556 | 23.82% | 66354 | 3.27% | 66111 | 2.89% | **65467** | **1.89%** |
| u2319 | 234256 | 240657 | 2.73% | 236054 | 0.77% | 237061 | 1.20% | 235667 | 0.60% | **234765** | **0.22%** | 236201 | 0.83% | 235093 | 0.36% |
| pcb3038 | 137694 | 142320 | 3.36% | 141418 | 2.70% | 142646 | 3.60% | 140351 | 1.93% | 139547 | 1.35% | 141446 | 2.72% | **139325** | **1.18%** |
| fl3795 | 28772 | 35138 | 22.13% | **33971** | **18.07%** | 36294 | 26.14% | 43940 | 52.72% | 36803 | 27.91% | 40183 | 39.66% | 35715 | 24.13% |
| pr2392 | 378032 | 384727 | 1.77% | 388518 | 2.77% | 386985 | 2.37% | 385057 | 1.86% | 385073 | 1.86% | 387623 | 2.54% | **380722** | **0.71%** |
| fnl4461 | 182566 | 187380 | 2.64% | 186985 | 2.42% | 187913 | 2.93% | 185869 | 1.81% | **184057** | **0.82%** | 186521 | 2.17% | 184776 | 1.21% |
| d2103 | 80450 | 83622 | 3.94% | 82614 | 2.69% | 83690 | 4.03% | 86119 | 7.05% | 83644 | 3.97% | 83360 | 3.62% | **81813** | **1.69%** |
| rl5934 | 556045 | 588550 | 5.85% | 579206 | 4.17% | 589806 | 6.07% | 843158 | 51.63% | **570853** | **2.66%** | 594357 | 6.89% | 574556 | 3.33% |
| rl5915 | 565530 | 589372 | 4.22% | 588542 | 4.07% | 585404 | 3.51% | 809375 | 43.12% | **578232** | **2.25%** | 584327 | 3.32% | 583477 | 3.17% |
| usa13509 | 19982859 | 20947758 | 4.83% | 20613997 | 3.16% | 21033416 | 5.26% | 28386893 | 42.06% | 21193246 | 6.06% | 20723480 | 3.71% | **20396752** | **2.07%** |
| brd14051 | 469385 | 492159 | 4.85% | 480186 | 2.30% | 489324 | 4.25% | 506961 | 8.01% | 485812 | 3.50% | 482790 | 2.86% | **479123** | **2.07%** |
| d18512 | 645238 | 672990 | 4.30% | 662312 | 2.65% | 667466 | 3.44% | 701169 | 8.67% | 663460 | 2.82% | 662022 | 2.60% | **656164** | **1.69%** |
| rl11849 | 923288 | 994084 | 7.67% | 955040 | 3.44% | 973842 | 5.48% | 1866653 | 102.17% | **948548** | **2.74%** | 953754 | 3.30% | 962460 | 4.24% |
| d15112 | 1573084 | 1659366 | 5.48% | 1613134 | 2.55% | 1631994 | 3.74% | 1978136 | 25.75% | 1614098 | 2.61% | 1612163 | 2.48% | **1598467** | **1.61%** |
| Average | - | 1934631 | 5.54% | 1902019 | 3.92% | 1936648 | 5.48% | 2589207 | 26.51% | 1941749 | 4.43% | 1911024 | 5.68% | **1883850** | **3.52%** |

Table 10: Results on small TSPLIB instances (with 0-500 nodes). The hyperparameter settings are the default settings as used by Fu et al. (2021).

| Instance | Optimal | Zero | | Att-GCN | | DIMES | | UTSP | | SoftDist | | DIFUSCO | | GT-Prior | |
|---|---|---|---|---|---|---|---|---|---|---|---|---|---|---|---|
| | | Length↓ | Gap↓ | Length↓ | Gap↓ | Length↓ | Gap↓ | Length↓ | Gap↓ | Length↓ | Gap↓ | Length↓ | Gap↓ | Length↓ | Gap↓ |
| st70 | 675 | **676** | **0.15%** | **676** | **0.15%** | 1056 | 56.44% | **676** | **0.15%** | 694 | 2.81% | **676** | **0.15%** | **676** | **0.15%** |
| kroA200 | 29368 | 29635 | 0.91% | **29368** | **0.00%** | 29464 | 0.33% | 29529 | 0.55% | 29383 | 0.05% | 29831 | 1.58% | 29397 | 0.10% |
| eil76 | 538 | **538** | **0.00%** | **538** | **0.00%** | 803 | 49.26% | **538** | **0.00%** | **538** | **0.00%** | **538** | **0.00%** | **538** | **0.00%** |
| pr144 | 58537 | 58554 | 0.03% | 67632 | 15.54% | 72458 | 23.78% | **58537** | **0.00%** | 66184 | 13.06% | 58901 | 0.62% | **58537** | **0.00%** |
| rat195 | 2323 | 2365 | 1.81% | **2323** | **0.00%** | 2331 | 0.34% | 2352 | 1.25% | **2323** | **0.00%** | 2337 | 0.60% | 2328 | 0.22% |
| eil51 | 426 | **427** | **0.23%** | **427** | **0.23%** | 653 | 53.29% | **427** | **0.23%** | **427** | **0.23%** | **427** | **0.23%** | **427** | **0.23%** |
| bier127 | 118282 | 118580 | 0.25% | **118282** | **0.00%** | 118715 | 0.37% | **118282** | **0.00%** | 118423 | 0.12% | 118657 | 0.32% | **118282** | **0.00%** |
| lin105 | 14379 | **14379** | **0.00%** | **14379** | **0.00%** | 16437 | 14.31% | **14379** | **0.00%** | 15073 | 4.83% | 14401 | 0.15% | **14379** | **0.00%** |
| kroD100 | 21294 | **21294** | **0.00%** | **21294** | **0.00%** | 28391 | 33.33% | 21309 | 0.07% | **21294** | **0.00%** | 21374 | 0.38% | **21294** | **0.00%** |
| pr152 | 73682 | 73880 | 0.27% | **73682** | **0.00%** | 86257 | 17.07% | **73682** | **0.00%** | **73682** | **0.00%** | 74029 | 0.47% | **73682** | **0.00%** |
| kroA100 | 21282 | **21282** | **0.00%** | **21282** | **0.00%** | 25168 | 18.26% | **21282** | **0.00%** | **21282** | **0.00%** | 21396 | 0.54% | **21282** | **0.00%** |
| ts225 | 126643 | 127147 | 0.40% | 126713 | 0.06% | 143360 | 13.20% | 126726 | 0.07% | 126962 | 0.25% | **126643** | **0.00%** | **126643** | **0.00%** |
| rd400 | 15281 | 15819 | 3.52% | 15413 | 0.86% | 15829 | 3.59% | 15418 | 1.96% | 15418 | 0.90% | **15350** | **0.45%** | 15454 | 1.13% |
| kroB100 | 22141 | 22193 | 0.23% | **22141** | **0.00%** | 26014 | 17.49% | **22141** | **0.00%** | **22141** | **0.00%** | 22601 | 2.08% | **22141** | **0.00%** |
| d198 | 15780 | 15883 | 0.65% | **15784** | **0.03%** | 16016 | 1.50% | 15874 | 0.60% | 15806 | 0.16% | 15859 | 0.50% | 15789 | 0.06% |
| eil101 | 629 | 630 | 0.16% | **629** | **0.00%** | 914 | 45.31% | **629** | **0.00%** | **629** | **0.00%** | **629** | **0.00%** | **629** | **0.00%** |
| linhp318 | 41345 | 43250 | 4.61% | 42359 | 2.45% | 43263 | 4.64% | 42453 | 2.68% | 43111 | 4.27% | 42336 | 2.40% | **42212** | **2.10%** |
| gil262 | 2378 | 2433 | 2.31% | 2383 | 0.21% | 2482 | 4.37% | 2394 | 0.67% | 2392 | 0.59% | **2380** | **0.08%** | 2389 | 0.46% |
| rat99 | 1211 | **1211** | **0.00%** | **1211** | **0.00%** | 1218 | 0.58% | **1211** | **0.00%** | **1211** | **0.00%** | 1214 | 0.25% | **1211** | **0.00%** |
| berlin52 | 7542 | **7542** | **0.00%** | **7542** | **0.00%** | 10569 | 40.14% | **7542** | **0.00%** | **7542** | **0.00%** | **7542** | **0.00%** | **7542** | **0.00%** |
| kroC100 | 20749 | **20749** | **0.00%** | **20749** | **0.00%** | 24666 | 18.88% | **20749** | **0.00%** | **20749** | **0.00%** | 20901 | 0.73% | **20749** | **0.00%** |
| pr226 | 80369 | 80822 | 0.56% | 83203 | 3.53% | 84543 | 5.19% | 81060 | 0.86% | 85411 | 6.27% | 83028 | 3.31% | **80369** | **0.00%** |
| fl417 | 11861 | 11932 | 0.60% | 12014 | 1.29% | 14036 | 18.34% | 45810 | 286.22% | 14897 | 25.60% | 13977 | 17.84% | **11907** | **0.39%** |
| kroE100 | 22068 | **22068** | **0.00%** | **22068** | **0.00%** | 26602 | 18.10% | **22068** | **0.00%** | **22068** | **0.00%** | 22135 | 0.30% | **22068** | **0.00%** |
| pr76 | 108159 | **108159** | **0.00%** | **108159** | **0.00%** | 130741 | 20.88% | **108159** | **0.00%** | 109325 | 1.08% | 111683 | 3.26% | **108159** | **0.00%** |
| ch130 | 6110 | 6149 | 0.64% | **6111** | **0.02%** | 7706 | 26.12% | 6120 | 0.16% | 6248 | 2.26% | 6157 | 0.77% | **6111** | **0.02%** |
| rd100 | 7910 | **7910** | **0.00%** | **7910** | **0.00%** | 14528 | 83.67% | **7910** | **0.00%** | 7932 | 0.28% | **7910** | **0.00%** | **7910** | **0.00%** |
| tsp225 | 3916 | 3982 | 1.69% | 3923 | 0.18% | 3945 | 0.74% | 3966 | 1.28% | **3919** | **0.08%** | 3920 | 0.10% | 3923 | 0.18% |
| pr264 | 49135 | 49552 | 0.85% | **49135** | **0.00%** | 49248 | 0.23% | 49844 | 1.44% | 49309 | 0.35% | 49180 | 0.09% | **49135** | **0.00%** |
| pr124 | 59030 | **59030** | **0.00%** | **59030** | **0.00%** | 76615 | 29.79% | **59030** | **0.00%** | 59524 | 0.84% | 59385 | 0.60% | **59030** | **0.00%** |
| kroA150 | 26524 | 26726 | 0.76% | **26525** | **0.00%** | 26719 | 0.74% | 26528 | 0.02% | **26525** | **0.00%** | 26556 | 0.12% | **26525** | **0.00%** |
| kroB200 | 29437 | 29619 | 0.62% | 29455 | 0.06% | 29511 | 0.25% | 29552 | 0.39% | **29438** | **0.00%** | 29659 | 0.75% | 29475 | 0.13% |
| kroB150 | 26130 | 26143 | 0.05% | 26132 | 0.01% | 26335 | 0.78% | 26176 | 0.18% | **26130** | **0.00%** | 26149 | 0.07% | **26130** | **0.00%** |
| pr107 | 44303 | 44358 | 0.12% | 44387 | 0.19% | 48621 | 9.75% | **44303** | **0.00%** | **44303** | **0.00%** | 44387 | 0.19% | **44303** | **0.00%** |
| lin318 | 42029 | 43250 | 2.91% | 42352 | 0.77% | 43116 | 2.59% | 42453 | 1.01% | 43111 | 2.57% | 42646 | 1.47% | **42212** | **0.44%** |
| pr136 | 96772 | 97515 | 0.77% | **96772** | **0.00%** | 119314 | 23.29% | **96772** | **0.00%** | **96772** | **0.00%** | 96781 | 0.01% | **96772** | **0.00%** |
| pr299 | 48191 | 48979 | 1.64% | 48280 | 0.18% | 48257 | 0.14% | 48594 | 0.84% | **48241** | **0.10%** | 48306 | 0.24% | 48303 | 0.23% |
| u159 | 42080 | **42080** | **0.00%** | **42080** | **0.00%** | 43188 | 2.63% | **42080** | **0.00%** | 42316 | 0.75% | 42685 | 1.44% | **42080** | **0.00%** |
| a280 | 2579 | 2633 | 2.09% | **2579** | **0.00%** | 2581 | 0.08% | 2589 | 0.39% | 2581 | 0.08% | **2579** | **0.00%** | 2585 | 0.23% |
| pr439 | 107217 | 109872 | 2.48% | 108631 | 1.32% | 108602 | 1.29% | 108424 | 1.13% | 115530 | 7.75% | 108855 | 1.53% | **107656** | **0.41%** |
| ch150 | 6528 | 6562 | 0.52% | **6528** | **0.00%** | 8178 | 25.28% | **6528** | **0.00%** | **6528** | **0.00%** | 6533 | 0.08% | **6528** | **0.00%** |
| d493 | 35002 | 35874 | 2.49% | **35373** | **1.06%** | 35522 | 1.49% | 36384 | 3.95% | 35480 | 1.37% | 35537 | 1.53% | 35487 | 1.39% |
| pcb442 | 50778 | 52292 | 2.98% | 51098 | 0.63% | 51147 | 0.73% | 51775 | 1.96% | 51177 | 0.79% | **50976** | **0.39%** | 51095 | 0.62% |
| Average | - | 35208 | 0.87% | 35268 | 0.67% | 38711 | 16.01% | 35870 | 7.16% | 35630 | 1.80% | 35280 | 1.06% | **34961** | **0.20%** |

# G  HYPERPARAMETER TUNING WITH SMAC3

In addition to the grid search method employed in the main content of this paper, we also conducted hyperparameter tuning using the Sequential Model-based Algorithm Configuration (SMAC3) framework (Lindauer et al., 2022). SMAC3 is designed for optimizing algorithm configurations through an efficient and adaptive search process that balances exploration and exploitation of the hyperparameter space.

The SMAC3 framework utilizes a surrogate model based on tree-structured Parzen estimators (TPE) to predict the performance of various hyperparameter configurations. This model is iteratively refined as configurations are evaluated, allowing SMAC3 to identify promising areas of the search space more effectively than traditional methods.

Table 11: Results on medium TSPLIB instances (with 500-2000 nodes). The hyperparameter settings are the default settings as used by Fu et al. (2021).

| Instance | Optimal | Zero | | Att-GCN | | DIMES | | UTSP | | SoftDist | | DIFUSCO | | GT-Prior | |
|---|---|---|---|---|---|---|---|---|---|---|---|---|---|---|---|
| | | Length ↓ | Gap ↓ | Length ↓ | Gap ↓ | Length ↓ | Gap ↓ | Length ↓ | Gap ↓ | Length ↓ | Gap ↓ | Length ↓ | Gap ↓ | Length ↓ | Gap ↓ |
| u574 | 36905 | 38171 | 3.43% | 37545 | 1.73% | 37803 | 2.43% | 38018 | 3.02% | 37545 | 1.73% | **37026** | **0.33%** | 37441 | 1.45% |
| pcb1173 | 56892 | 60231 | 5.87% | 58452 | 2.74% | 58664 | 3.11% | 59761 | 5.04% | 58209 | 2.31% | **57717** | **1.45%** | 58251 | 2.39% |
| u1432 | 152970 | 162741 | 6.39% | 157322 | 2.85% | 157056 | 2.67% | 159654 | 4.37% | 155566 | 1.70% | **154734** | **1.15%** | 156126 | 2.06% |
| rat783 | 8806 | 9230 | 4.81% | 8995 | 2.15% | 9088 | 3.20% | 9124 | 3.61% | 8936 | 1.48% | **8863** | **0.65%** | 8986 | 2.04% |
| fl1400 | 20127 | **20917** | **3.93%** | 23347 | 16.00% | 20932 | 4.00% | 37919 | 88.40% | 30111 | 49.61% | 22608 | 12.33% | 21272 | 5.69% |
| vm1084 | 239297 | 251602 | 5.14% | 242848 | 1.48% | 245994 | 2.80% | 252204 | 5.39% | 243541 | 1.77% | **242375** | **1.29%** | 244267 | 2.08% |
| rat575 | 6773 | 6982 | 3.09% | 6901 | 1.89% | 7053 | 4.13% | 6959 | 2.75% | 6871 | 1.45% | **6801** | **0.41%** | 6842 | 1.02% |
| vm1748 | 336556 | 352556 | 4.75% | 344077 | 2.23% | 347356 | 3.21% | 372117 | 10.57% | 344193 | 2.27% | **340888** | **1.29%** | 343973 | 2.20% |
| rl1889 | 316536 | 335641 | 6.04% | 325270 | 2.76% | 338164 | 6.83% | 358570 | 13.28% | 329839 | 4.20% | **322969** | **2.03%** | 328399 | 3.75% |
| u724 | 41910 | 43487 | 3.76% | 42525 | 1.47% | 42915 | 2.40% | 43106 | 2.85% | 42508 | 1.43% | **42081** | **0.41%** | 42420 | 1.22% |
| d1291 | 50801 | 52757 | 3.85% | 52063 | 2.48% | 53833 | 5.97% | 54231 | 6.75% | 52230 | 2.81% | **51937** | **2.24%** | 52553 | 3.45% |
| pr1002 | 259045 | 273143 | 5.44% | 264647 | 2.16% | 267949 | 3.44% | 268931 | 3.82% | 266468 | 2.87% | **263242** | **1.62%** | 264704 | 2.18% |
| fl1577 | 22249 | **23351** | **4.95%** | 26082 | 17.23% | 23954 | 7.66% | 27592 | 24.01% | 28630 | 28.68% | 25493 | 14.58% | 27531 | 23.74% |
| nrw1379 | 56638 | 58991 | 4.15% | 57681 | 1.84% | 57737 | 1.94% | 65399 | 15.47% | 58021 | 2.44% | **57297** | **1.16%** | 57654 | 1.79% |
| rl1304 | 252948 | 270179 | 6.81% | 259681 | 2.66% | 270057 | 6.76% | 268425 | 6.12% | 264884 | 4.72% | **255970** | **1.19%** | 263748 | 4.27% |
| d657 | 48912 | 50971 | 4.21% | 49798 | 1.81% | 50577 | 3.40% | 50437 | 3.12% | 49657 | 1.52% | **49153** | **0.49%** | 49616 | 1.44% |
| p654 | 34643 | **35266** | **1.80%** | 36233 | 4.59% | 35873 | 3.55% | 49921 | 44.10% | 44016 | 27.06% | 37936 | 9.51% | 35979 | 3.86% |
| d1655 | 62128 | 66819 | 7.55% | 63970 | 2.96% | 64668 | 4.09% | 75875 | 22.13% | 64467 | 3.76% | **63575** | **2.33%** | 63610 | 2.39% |
| u1817 | 57201 | 61671 | 7.81% | 59226 | 3.54% | 60219 | 5.28% | 63152 | 10.40% | 59585 | 4.17% | **58780** | **2.76%** | 59318 | 3.70% |
| u1060 | 224094 | 232616 | 3.80% | **227340** | **1.45%** | 232619 | 3.80% | 236167 | 5.39% | 228869 | 2.13% | 227868 | 1.68% | 229515 | 2.42% |
| rl1323 | 270199 | 283701 | 5.00% | 276363 | 2.28% | 282500 | 4.55% | 282676 | 4.62% | 278379 | 3.03% | **274038** | **1.42%** | 278283 | 2.99% |
| Average | - | 128143 | 4.88% | 124779 | 3.73% | 126905 | 4.06% | 132392 | 13.58% | 126310 | 7.20% | **123873** | **2.87%** | 125261 | 3.63% |

Table 12: Results on large TSPLIB instances (with more than 2000 nodes). The hyperparameter settings are the default settings as used by Fu et al. (2021).

| Instance | Optimal | Zero | | Att-GCN | | DIMES | | UTSP | | SoftDist | | DIFUSCO | | GT-Prior | |
|---|---|---|---|---|---|---|---|---|---|---|---|---|---|---|---|
| | | Length ↓ | Gap ↓ | Length ↓ | Gap ↓ | Length ↓ | Gap ↓ | Length ↓ | Gap ↓ | Length ↓ | Gap ↓ | Length ↓ | Gap ↓ | Length ↓ | Gap ↓ |
| u2152 | 64253 | 68293 | 6.29% | 66717 | 3.83% | 69322 | 7.89% | 71240 | 10.87% | 96834 | 50.71% | 77826 | 21.12% | **66600** | **3.65%** |
| u2319 | 234256 | 243093 | 3.77% | 237114 | 1.22% | 251125 | 7.20% | 244142 | 4.22% | **235644** | **0.59%** | 237035 | 1.19% | 236159 | 0.81% |
| pcb3038 | 137694 | 150518 | 9.31% | 142015 | 3.14% | 163500 | 18.74% | 148143 | 7.59% | 141977 | 3.11% | 157341 | 14.27% | **141372** | **2.67%** |
| fl3795 | 28772 | **30032** | **4.38%** | 35694 | 24.06% | 35201 | 22.34% | 50835 | 76.68% | 36579 | 27.13% | 42120 | 46.39% | 38852 | 35.03% |
| pr2392 | 378032 | 392998 | 3.96% | 391367 | 3.53% | 426194 | 12.74% | 401216 | 6.13% | 438424 | 15.98% | 430218 | 13.80% | **385009** | **1.85%** |
| fnl4461 | 182566 | 192471 | 5.43% | 187802 | 2.87% | 235876 | 29.20% | 229934 | 25.95% | 186632 | 2.23% | 192868 | 5.64% | **186359** | **2.08%** |
| d2103 | 80450 | 88698 | 10.25% | 83881 | 4.26% | 96968 | 20.53% | 88022 | 9.41% | 84662 | 5.24% | 90773 | 12.83% | **82723** | **2.83%** |
| rl5934 | 556045 | 590393 | 6.18% | **576829** | **3.74%** | 703750 | 26.56% | 781490 | 40.54% | 647689 | 16.48% | 645291 | 16.05% | 592889 | 6.63% |
| rl5915 | 565530 | 603653 | 6.74% | **587231** | **3.84%** | 694199 | 22.75% | 809014 | 43.05% | 644676 | 14.00% | 656872 | 16.15% | 591517 | 4.60% |
| usa13509 | 19982859 | 21177174 | 5.98% | **20733868** | **3.76%** | 442759283 | 2115.70% | 1115269461 | 5481.13% | 21094456 | 5.56% | 22241850 | 11.30% | 20742301 | 3.80% |
| brd14051 | 469385 | 496359 | 5.75% | 484032 | 3.12% | 3757018 | 700.40% | 13600052 | 2797.42% | 493461 | 5.13% | 489311 | 4.25% | **483657** | **3.04%** |
| d18512 | 645238 | 685983 | 6.31% | 665993 | 3.22% | 4922388 | 662.88% | 22893796 | 3448.12% | 664334 | 2.96% | 663087 | 2.77% | **659537** | **2.22%** |
| rl11849 | 923288 | 1014118 | 9.84% | **961746** | **4.17%** | 7381138 | 699.44% | 40891587 | 4328.91% | 990268 | 7.25% | 977396 | 5.86% | 970070 | 5.07% |
| d15112 | 1573084 | 1681649 | 6.90% | 1621028 | 3.05% | 19507797 | 1140.10% | 71782581 | 4463.18% | **1615421** | **2.69%** | 1653223 | 5.09% | 1618636 | 2.90% |
| Average | - | 1958245 | 6.51% | **1912522** | **4.84%** | 34357411 | 391.89% | 90518679 | 1481.66% | 1955075 | 11.36% | 2039657 | 12.62% | 1913977 | 5.51% |

For our experiments, we configured SMAC3 to optimize the same hyperparameters as those previously tuned via grid search. The search space remains identical to that demonstrated in Table 1, However, we set SMAC3 to search for 50 epochs (50 different hyperparameter combinations) instead of exploring the entire search space (864 different combinations) and the time limit for MCTS was set to 50 seconds for TSP-500, 100 seconds for TSP-1000, and 1000 seconds for TSP-10000. We show the time cost of each tuning method in Table 13.

The results of these experiments, including the hyperparameter settings identified by SMAC3 and their corresponding performance metrics, are presented in Tables 14 and 15. As shown, the performance achieved by SMAC3 is comparable to that of grid search. Specifically, for TSP-500 and TSP-1000, SMAC3 produces results similar to those of Att-GCN DIFUSCO and GT-Prior, with even better outcomes observed on TSP-10000. This improvement can

Table 13: The Comparison of Tuning Time Between Grid Search and SMAC3. "h" indicates hours.

| | Grid Search | SMAC3 |
|---|---|---|
| TSP-500 | 24h | 1.39h |
| TSP-1000 | 48h | 2.78h |
| TSP-10000 | 6h | 3.47h |

be attributed to the extended tuning time allowed by SMAC3 compared to grid search. Given the significant difference in time costs, SMAC3 proves to be an efficient and economical option for tuning MCTS hyperparameters.

## H  GT-PRIOR INFORMATION

We provide detailed information about GT-Prior for constructing the heatmap for TSP500, TSP1000, and TSP10000 as follows:

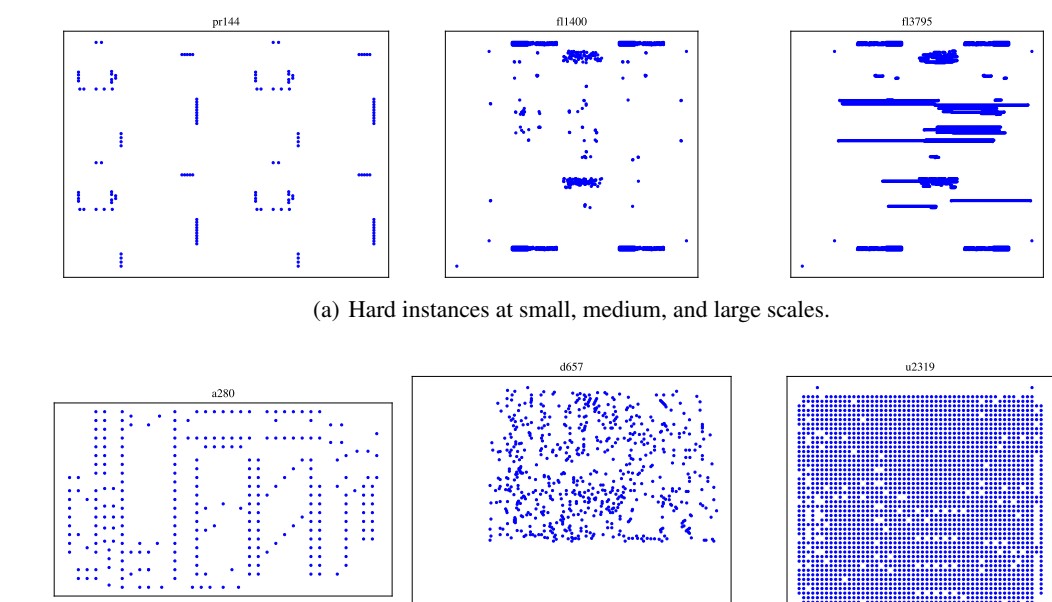

(a) Hard instances at small, medium, and large scales.

(b) Easy instances at small, medium, and large scales.

Figure 6: Representative TSPLIB instances visualization.

Table 14: Tune parameters of all the methods for TSP500, TSP1000 and TSP10000 by grid search (the left table) and SMAC3 (the right table).

| | METHOD | ALPHA | BETA | H | MCN | UH | MD | | METHOD | ALPHA | BETA | H | MCN | UH | MD |
|---|---|---|---|---|---|---|---|---|---|---|---|---|---|---|---|
| | ZERO | 2 | 10 | 2 | 5 | 0 | 100 | | ZERO | 0 | 150 | 2 | 5 | 0 | 50 |
| | ATT-GCN | 0 | 150 | 5 | 5 | 0 | 100 | | ATT-GCN | 2 | 150 | 2 | 5 | 0 | 100 |
| | DIMES | 0 | 100 | 5 | 5 | 0 | 200 | | DIMES | 0 | 100 | 5 | 5 | 0 | 200 |
| TSP500 | DIFUSCO | 1 | 150 | 2 | 5 | 0 | 50 | TSP500 | DIFUSCO | 1 | 150 | 2 | 5 | 0 | 50 |
| | UTSP | 0 | 100 | 5 | 5 | 0 | 50 | | UTSP | 0 | 100 | 5 | 5 | 0 | 50 |
| | SOFTDIST | 1 | 100 | 5 | 20 | 0 | 200 | | SOFTDIST | 1 | 100 | 5 | 20 | 0 | 200 |
| | GT-PRIOR | 0 | 10 | 5 | 5 | 1 | 200 | | GT-PRIOR | 0 | 10 | 5 | 5 | 1 | 200 |
| | ZERO | 1 | 100 | 5 | 5 | 0 | 100 | | ZERO | 0 | 150 | 2 | 5 | 0 | 100 |
| | ATT-GCN | 0 | 150 | 5 | 5 | 0 | 200 | | ATT-GCN | 2 | 150 | 2 | 5 | 0 | 100 |
| | DIMES | 0 | 150 | 2 | 5 | 0 | 200 | | DIMES | 2 | 150 | 5 | 5 | 0 | 100 |
| TSP1000 | DIFUSCO | 0 | 150 | 2 | 5 | 1 | 200 | TSP1000 | DIFUSCO | 0 | 150 | 2 | 5 | 1 | 200 |
| | UTSP | 1 | 100 | 5 | 5 | 0 | 50 | | UTSP | 0 | 100 | 5 | 5 | 0 | 50 |
| | SOFTDIST | 0 | 150 | 2 | 20 | 1 | 200 | | SOFTDIST | 1 | 100 | 2 | 50 | 1 | 200 |
| | GT-PRIOR | 1 | 10 | 5 | 5 | 1 | 200 | | GT-PRIOR | 0 | 150 | 2 | 5 | 1 | 200 |
| | ZERO | 0 | 100 | 2 | 20 | 0 | 10 | | ZERO | 0 | 100 | 2 | 20 | 0 | 10 |
| | ATT-GCN | 1 | 150 | 2 | 5 | 1 | 50 | | ATT-GCN | 1 | 150 | 2 | 5 | 1 | 50 |
| TSP10000 | DIMES | 1 | 100 | 2 | 20 | 0 | 10 | TSP10000 | DIMES | 1 | 100 | 2 | 20 | 0 | 10 |
| | DIFUSCO | 0 | 100 | 5 | 20 | 0 | 50 | | DIFUSCO | 0 | 100 | 5 | 20 | 0 | 50 |
| | SOFTDIST | 2 | 100 | 5 | 20 | 0 | 10 | | SOFTDIST | 2 | 100 | 5 | 20 | 0 | 10 |
| | GT-PRIOR | 1 | 100 | 10 | 1000 | 1 | 100 | | GT-PRIOR | 1 | 100 | 10 | 1000 | 1 | 100 |

```
# GT_Prior
# TSP500:
[4.40078125e-01, 2.56265625e-01, 1.32750000e-01, 7.32656250e-02,
4.08125000e-02, 2.35937500e-02, 1.34062500e-02, 7.75000000e-03,
4.48437500e-03, 2.73437500e-03, 1.78125000e-03, 1.18750000e-03,
6.87500000e-04, 3.75000000e-04, 3.75000000e-04, 1.87500000e-04,
7.81250000e-05, 1.56250000e-05, 4.68750000e-05, 1.56250000e-05,
4.68750000e-05, 3.12500000e-05, 1.56250000e-05, 1.56250000e-05]

# TSP1000:
[4.37554687e-01, 2.54718750e-01, 1.37671875e-01, 7.41093750e-02,
3.97890625e-02, 2.35156250e-02, 1.32265625e-02, 7.45312500e-03,
4.73437500e-03, 3.00781250e-03, 1.59375000e-03, 1.08593750e-03,
5.62500000e-04, 2.96875000e-04, 2.65625000e-04, 1.71875000e-04,
```

Table 15: Results of Hyperparameter Tuning using SMAC3. The underlined figures in the table indicate results that are equal to or better than those of Grid Search, rounded to two decimal places.

| METHOD | TYPE | TSP-500 LENGTH ↓ | TSP-500 GAP ↓ | TSP-500 TIME ↓ | TSP-1000 LENGTH ↓ | TSP-1000 GAP ↓ | TSP-1000 TIME ↓ | TSP-10000 LENGTH ↓ | TSP-10000 GAP ↓ | TSP-10000 TIME ↓ |
|---|---|---|---|---|---|---|---|---|---|---|
| CONCORDE | OR(EXACT) | 16.55* | — | 37.66M | 23.12* | — | 6.65H | N/A | N/A | N/A |
| GUROBI | OR(EXACT) | 16.55 | 0.00% | 45.63H | N/A | N/A | N/A | N/A | N/A | N/A |
| LKH-3 (DEFAULT) | OR | 16.55 | 0.00% | 46.28M | 23.12 | 0.00% | 2.57H | 71.78* | — | 8.8H |
| LKH-3 (LESS TRAILS) | OR | 16.55 | 0.00% | 3.03M | 23.12 | 0.00% | 7.73M | 71.79 | — | 51.27M |
| ZERO | MCTS | 16.67 | 0.73% | **0.00M+** **1.67M** | 23.39 | 1.17% | **0.00M+** **3.34M** | 74.44 | 3.71% | **0.00M+** **16.78M** |
| ATT-GCN† | SL+MCTS | 16.66 | 0.69% | 0.52M+ 1.67M | 23.38 | 1.15% | 0.73M+ 3.34M | 73.87 | 2.92% | 4.16M+ 16.77M |
| DIMES† | RL+MCTS | 16.67 | 0.73% | 0.97M+ 1.67M | 23.42 | 1.31% | 2.08M+ 3.34M | 74.17 | 3.33% | 4.65M+ 16.77M |
| UTSP† | UL+MCTS | 16.72 | 1.07% | 1.37M+ 1.67M | 23.51 | 1.68% | 3.35M+ 3.34M | — | — | — |
| SOFTDIST† | SOFTDIST+MCTS | 16.62 | 0.46% | **0.00M+** **1.67M** | 23.33 | 0.90% | **0.00M+** **3.34M** | 75.34 | 4.97% | **0.00M+** **16.78M** |
| DIFUSCO† | SL+MCTS | **16.62** | **0.43%** | 3.61M+ 1.67M | **23.24** | **0.53%** | 11.86M+ 3.34M | 73.26 | 2.06% | 28.51M+ 16.78M |
| GT-PRIOR | PRIOR+MCTS | 16.63 | 0.50% | **0.00M+** **1.67M** | 23.32 | 0.85% | **0.00M+** **3.34M** | 73.26 | 2.07% | **0.00M+** **16.78M** |

```
1.01562500e-04, 4.68750000e-05, 1.56250000e-05, 3.12500000e-05,
2.34375000e-05, 7.81250000e-06, 1.56250000e-05]

# TSP10000:
[4.4175625e-01, 2.5409375e-01, 1.3292500e-01, 7.1950000e-02,
3.9518750e-02, 2.3750000e-02, 1.4143750e-02, 8.0937500e-03,
4.9125000e-03, 3.3312500e-03, 1.8437500e-03, 1.1125000e-03,
8.3750000e-04, 5.5625000e-04, 3.7500000e-04, 2.6250000e-04,
1.8125000e-04, 8.7500000e-05, 6.8750000e-05, 5.0000000e-05,
5.0000000e-05, 2.5000000e-05, 2.5000000e-05, 6.2500000e-06,
1.2500000e-05, 6.2500000e-06, 6.2500000e-06, 6.2500000e-06,
6.2500000e-06, 6.2500000e-06]
```

