# OpenReview forum: "Rethinking the "Heatmap + Monte Carlo Tree Search'' Paradigm for Solving Large Scale TSP"
_ICLR.cc/2025/Conference — Submitted to ICLR 2025_

### Official Review · Reviewer_3WVZ · 2024-10-25

**Soundness:** 2
**Presentation:** 2
**Contribution:** 1
**Rating:** 3
**Confidence:** 4

**Summary:**

This paper emphasizes the importance of tuning the hyperparameters of MCTS in the heatmap+MCTS strategy for solving TSP. It conducts experiments to analyze the impact of each hyperparameter setting in MCTS. Additionally, it proposes a method to generate the heatmap at a very low computational cost and verifies its performance through experiments.

**Strengths:**

- This paper experimentally shows that significant performance improvements can be achieved in solving TSP using heatmap+MCTS strategy by tuning the hyperparameters of MCTS.

- The proposed GT-Prior method demonstrates relatively superior results while consuming less computational cost compared to existing methods.

**Weaknesses:**

- One major part of this paper, the hyperparameter tuning of MCTS, lacks novelty. In this paper, the authors simply tuned the existing hyperparameters of MCTS without introducing new hyperparameters or proposing a new tuning method, and in doing so, confirmed that it is possible to improve the accuracy of solving TSP. The authors claim that previous studies have not paid attention to the values of MCTS hyperparameters and emphasize the need for careful tuning of MCTS hyperparameters. In previous studies proposing new heatmap generation methods, it seems that MCTS hyperparameters were intentionally left untuned to allow for a fair comparison of the heatmap's performance. While this paper’s confirmation that tuning MCTS hyperparameters can improve TSP solution accuracy has some value, it is not novel that tuning hyperparameters can enhance solver performance. Therefore, simply finding better values for existing hyperparameters through traditional tuning methods is considered to lack novelty.


- The newly proposed GT-Prior heatmap generation method in this paper suggests a way to create a heatmap with minimal computation. Although this method showed the best performance in TSP-10000, it yielded worse results than existing methods (DIFUSCO, SOFTDIST) in TSP-500 and TSP-1000. Additionally, since the GT-Prior method statistically calculates the likelihood of inclusion in the optimal solution based on a ranking of **distances between nodes**, it is likely difficult to apply this method to problems other than TSP, especially those with complex constraints.

**Questions:**

- In line 360, it is mentioned that 3,000 optimal solutions were used to calculate $\hat{P}_N()$ for TSP-500 and TSP-1000, while only 128 optimal solutions were used for TSP-10000. Given that TSP-10000 is larger in scale, I would expect more instances to be used than for the smaller TSP problems, yet only 128 were used. What is the reason for this? Similarly, I would like to know the authors' opinion on whether it is appropriate to calculate $\hat{P}_N()$ based on statistics from only 128 instances for TSP-10000.
- Additionally, please confirm whether the $\hat{P}_N()$ used in the experiments in Table 2 and Table 3 was generated using the same number of instances.

---

> ### Author Response · Authors · 2024-11-19
> **Response to Reviewer 3WVZ (Part 1 of 2)**
>
> Dear Reviewer 3WVZ,
>
> ### Opening Statement
>
> First of all, We earnestly request that you reconsider our work from a different  perspective. Our primary contribution does not lie in proposing a novel  hyperparameter tuning method; rather, we present a critical viewpoint  that challenges the current research trajectory in this field. Our work  aims to provoke a fundamental reassessment of the "Heatmap + MCTS"  paradigm for large-scale TSP solving. We firmly believe that our  insights and conclusions make a significant contribution by encouraging  the research community to critically examine established assumptions and  potentially redirect research efforts more effectively.
>
>
>
> > W1: **Lack of Novelty in Hyperparameter Tuning**: The focus on hyperparameter tuning for MCTS lacks novelty, as it involves tuning existing MCTS hyperparameters without introducing new ones or proposing new tuning methods. Previous studies intentionally left MCTS hyperparameters untuned to ensure fair heatmap performance comparisons, and this paper’s tuning approach lacks originality.
>
> We maintain that comparing the best achievable performance of each  heatmap approach provides the most fair evaluation, which motivated our  decision to employ naive grid search for MCTS hyperparameter tuning.
>
> For the novelty and contraibution of our work, we kindly direct you to the **Opening Statement** and the first point under "Common Concerns" in the Global  Response: "1. Regarding Novelty and Research Values".
>
>
>
> > W2: **Limited Applicability and Mixed Performance of GT-Prior**
>
> We would like to emphasize again that our primary objective is not to propose another SOTA heatmap. Instead, our work makes two critical observations:
>
> 1. From a problem-solving perspective, current heatmap methods still have significant room for improvement through proper MCTS hyperparameter tuning, as demonstrated in Table 2: Performance Improvement after Hyperparameter Tuning.
>
> 2. A simple, parameter-free heatmap, when combined with well-tuned MCTS parameters, can achieve comparable performance to previous resource-intensive methods.
>
> We believe these findings should prompt serious reflection within the research community, potentially leading to more efficient approaches for solving large-scale TSP instances. While we acknowledge that our GT-Prior and MCTS configurations are specifically designed for TSP's problem structure and therefore do not generalize to other types of combinatorial optimization problems, this limitation is inherent and, more importantly, falls outside the scope of our research objectives.
>
> We also kindly suggest the reviewer to refer to the second point under "Common Concerns" in the Global Response: "2. Generalization to Other CO Tasks"

---

> ### Author Response · Authors · 2024-11-19
> **Response to Reviewer 3WVZ (Part 2 of 2)**
>
> > Q1: **Use of Optimal Solutions for TSP-10000**: Line 360 mentions using 3,000 optimal solutions for TSP-500 and TSP-1000 but only 128 for TSP-10000. Given the larger scale of TSP-10000, why were fewer instances used? Additionally, is it appropriate to base statistics on only 128 instances for TSP-10000?
>
> The use of 128 instances for statistical collection was due to the substantial computational cost of solving TSP10k instances. To address this concern, we conducted additional experiments using 3,000 TSP10k instances to collect GT-Prior statistics. The results are as follows:
>
> GT-Prior from 3,000 instances:
>
> ```python
> [4.40954829e-01, 2.53630246e-01, 1.33515263e-01, 7.21532501e-02,
>  4.04198735e-02, 2.33079141e-02, 1.37615144e-02, 8.31309549e-03,
>  5.06598423e-03, 3.14667758e-03, 1.97710685e-03, 1.26587106e-03,
>  8.18402837e-04, 5.30468506e-04, 3.56334569e-04, 2.46000853e-04,
>  1.61667227e-04, 1.05600366e-04, 7.76336025e-05, 5.29001834e-05,
>  3.85334669e-05, 2.70667605e-05, 1.99000690e-05, 1.42333827e-05,
>  1.16000402e-05, 7.56669290e-06, 5.73335321e-06, 3.96668042e-06,
>  3.50001213e-06, 2.10000728e-06, 1.50000520e-06, 1.46667175e-06,1.16667071e-06, 1.03333692e-06]
> ```
>
> Compared to GT-Prior from 128 instances:
>
> ```python
> [4.4175625e-01, 2.5409375e-01, 1.3292500e-01, 7.1950000e-02,
>  3.9518750e-02, 2.3750000e-02, 1.4143750e-02, 8.0937500e-03,
>  4.9125000e-03, 3.3312500e-03, 1.8437500e-03, 1.1125000e-03,
>  8.3750000e-04, 5.5625000e-04, 3.7500000e-04, 2.6250000e-04,
>  1.8125000e-04, 8.7500000e-05, 6.8750000e-05, 5.0000000e-05,
>  5.0000000e-05, 2.5000000e-05, 2.5000000e-05, 6.2500000e-06, 1.2500000e-05,
>  6.2500000e-06, 6.2500000e-06, 6.2500000e-06, 6.2500000e-06, 6.2500000e-06]
> ```
>
> The comparison shows minimal differences in the statistical patterns (MSE: 8.5e-08, KL divergence: 5.68e-05). This is further validated by the performance on the TSP10k test set:
>
> |                           | Objective | Gap   |
> | ------------------------- | --------- | ----- |
> | GT-Prior (128 instances)  | 73.31     | 2.13% |
> | GT-Prior (3000 instances) | 73.25     | 2.05% |
>
> While the GT-Prior collected from more instances indeed achieves slightly better results, the improvement is marginal. Although we acknowledge that more instances would provide better statistical reliability, these results demonstrate that even a smaller number of instances can yield effective performance.
>
> This finding reinforces two key advantages of our proposed GT-Prior for large-scale TSP solving:
>
> 1. Scale-invariance: Prior knowledge collected from smaller instances (e.g., TSP500) can be effectively applied to larger instances (e.g., TSP10k)
>
> 2. Sample-efficiency: The method's performance is relatively insensitive to the number of instances used for collecting statistics, enabling cost-effective GT-Prior generation.
>
>
>
> > Q2: **Consistency in Instance Use Across Experiments**: Please confirm whether the heatmaps used in experiments (Table 2 and Table 3) were generated using the same number of instances.
>
> Yes, we maintained consistency in our experimental setup. For Tables 2 and 3, we used the same test instances as provided by [1] and the same heatmaps as provided by [2], ensuring fair comparison with previous works. This standardized evaluation protocol allows direct performance comparisons across different methods.
>
> Ref:
>
> [1] Zhang-Hua Fu, Kai-Bin Qiu, and Hongyuan Zha. Generalize a small pre-trained model to arbitrarily large tsp instances. In Proceedings of the AAAI conference on artificial intelligence, volume 35, pp. 7474–7482, 2021.
>
> [2] Yifan Xia, Xianliang Yang, Zichuan Liu, Zhihao Liu, Lei Song, and Jiang Bian. Position: Rethinking post-hoc search-based neural approaches for solving large-scale traveling salesman problems. In Proceedings of the 41st International Conference on Machine Learning, pp. 54178–54190, 2024.

---

> > ### Comment · Reviewer_3WVZ · 2024-11-27
> >
> > Thank you for your detailed response and additional explanations. I do not agree with the part where the importance of MCTS in the heatmap+MCTS method was overlooked. Fu et al. [1] proposed an excellent MCTS algorithm, and Qiu et al. [2] demonstrated in their paper that not using MCTS results in a significant drop in accuracy. Furthermore, the heatmap method presented in this paper is an algorithm that can only be used for the TSP, yet it did not show better performance than existing methods in TSP. I have reconsidered the contributions claimed by the authors and the value of this paper, but I have decided to maintain my original score.
> >
> > [1] Zhang-Hua Fu, Kai-Bin Qiu, and Hongyuan Zha. Generalize a small pre-trained model to arbitrarily large tsp instances. In Proceedings of the AAAI conference on artificial intelligence, volume 35, pp. 7474–7482, 2021.
> >
> > [2] Ruizhong Qiu, Zhiqing Sun, and Yiming Yang. Dimes: A differentiable meta solver for combinatorial optimization problems. Advances in Neural Information Processing Systems, 35:25531–25546, 2022.

---

> > > ### Author Response · Authors · 2024-11-27
> > >
> > > Thank you for your response.  We believe that reviewer 3WVZ and we have a fundamental discrepancy in viewing "**why we put persistent effort on 'heatmap + MCTS' for large-scale TSP**". Once again, our motivation is to **better solve such TSP instances**, in a more efficient and straightforward way, rather than designing sophisticated and fancy heatmap generation methods that cannot even push a significant margin. Aligning with the **Main Motivation**, we believe that the critical point we discussed and implemented about the importance of tuning MCTS, stands for a substantial advance to **Better Solve Large-scale TSP**. Further, we believe *current trends on solely designing heatmap diverges from the **Main Motivation**, and is misleading and harmful to the community, lacking diversity*. We respond to your specific points as follows.
> > >
> > > - First, regarding your statement,
> > > > "I do not agree with the part where the importance of MCTS in the heatmap+MCTS method was overlooked,"
> > >
> > >     We suppose there may either be **a factual misunderstanding or an intentional conceptual shift** in your argument. Your interpretation of *"the importance of MCTS"*  seems to focus on the binary notion of whether to use MCTS or not, whereas our emphasis on *"the importance of MCTS"*  specifically examines the impact of **deeper investigation to MCTS module** for solving large-scale TSP. We believe what we are trying to emphasize is clear and can be readily concluded. All prior works primarily concentrated on the construction of heatmaps and largely overlooked how **tuning the MCTS parameters could fundamentally enhance the performance** of the overall framework. This is a critical point we aim to highlight, as stated in our manuscript, as well as the rebuttal.
> > >
> > > - Second, the purpose of proposing GT-prior was not to achieve SOTA performance, but rather, as we claimed in the manuscript, to demonstrate that **even a simple and straightforward heatmap, which neither requires any model training nor parameter tuning, can achieve comparable outcomes** to those of previously elaborate heatmaps. *This in turn can support our suspicion of the efficacy of several fancy heatmap generation methods.* Furthermore, many of these heatmap designs often demand significant computational resources, which poses a challenge for efficiently solving large-scale TSP  problems. In this sense, **shouldn't we rethink what effort to make to approach the Main Motivation?**
> > >
> > > Based on your response, it seems that the main argument and core ideas of our paper may not have been fully understood. We look forward to your reply and engaging in a more in-depth exchange. Thank you again for your time and consideration.
> > >
> > > The Authors.

---

### Official Review · Reviewer_DdHK · 2024-11-04

**Soundness:** 3
**Presentation:** 2
**Contribution:** 2
**Rating:** 5
**Confidence:** 4

**Summary:**

The paper revisits the “heatmap + Monte Carlo Tree Search (MCTS)” approach for the Travelling Salesman Problem (TSP). It argues that excessive focus on heatmap refinement in previous research has overshadowed the significant impact of MCTS configuration. Through empirical evaluation, the authors demonstrate that tuning MCTS parameters is critical for solution quality, and a basic k-nearest neighbor-based heatmap performs comparably to complex models.

**Strengths:**

1, The paper highlights the often-overlooked role of MCTS in improving solution quality, shedding light on its critical impact in the TSP solution process.

2, By introducing a simple, training-free k-nearest neighbor-based heatmap algorithm, the paper demonstrates a practical approach that achieves performance comparable to state-of-the-art models.

3, Experiments across varying TSP scales reinforce the findings, with clear evidence supporting the role of MCTS tuning.

**Weaknesses:**

1, It would have been beneficial if the paper addressed a broader range of combinatorial optimization (CO) problems rather than focusing solely on TSP.

2, MCTS is a very powerful algorithm, and previous studies have shown significant performance improvements when MCTS is added. Considering this, the paper’s finding that ‘MCTS has a large impact and careful tuning improves performance’ doesn’t feel particularly novel. It would be helpful to include a discussion on the advantages of MCTS over other local search algorithms (e.g., 2-opt) to clarify why MCTS is chosen. Additionally, an analysis of the synergy between MCTS and the heatmap approach would add value, especially considering that, as seen in Table 2, algorithms like Dimes and UTSP perform worse than Zero when MCTS is used.

3, MCTS involves numerous hyperparameters and appears highly sensitive to these settings, which could make practical implementation challenging and less accessible.

**Questions:**

1, The paper currently addresses only the TSP problem. Do you anticipate that similar results would be achieved if this approach were applied to other CO tasks, such as CVRP? Or are these findings specific to TSP? I’m curious to know if the method generalizes well to tasks beyond TSP.

2, The experimental setup in Section 5.3 (“GENERALIZATION ABILITY”) could be clarified. For example, for DIMES on TSP-1000, it is unclear whether the original (ORI.) setup uses a model and MCTS tuned for TSP-500 or TSP-1000. Similarly, does the generalization (GEN.) setting use MCTS tuned specifically for TSP-1000, or is it using MCTS tuned for TSP-500? Adding a table or flowchart to illustrate the training and testing configurations, specifying the problem sizes for model training, MCTS tuning, and evaluation, would greatly enhance clarity.

---

> ### Author Response · Authors · 2024-11-19
> **Response to Reviewer DdHK (Part 1 of 2)**
>
> Dear Reviewer DdHK,
>
> ### Opening Statement
>
> First of all, We earnestly request that you reconsider our work from a different  perspective. Our primary contribution does not lie in proposing a novel  hyperparameter tuning method; rather, we present a critical viewpoint  that challenges the current research trajectory in this field. Our work  aims to provoke a fundamental reassessment of the "Heatmap + MCTS"  paradigm for large-scale TSP solving. We firmly believe that our  insights and conclusions make a significant contribution by encouraging  the research community to critically examine established assumptions and  potentially redirect research efforts more effectively.
>
>
>
> > W1: **Limited Scope to TSP**: The study focuses only on TSP, which limits the generality of its findings. Addressing other CO problems would have increased the paper's impact.
>
> Please refer to the second point under "Common Concerns" in the Global Response: "2. Generalization to Other CO Tasks"
>
>
>
> > W2: **Lack of Novelty in MCTS Contribution**: While the paper highlights the impact of MCTS tuning, this may not be seen as particularly novel, given the well-known power of MCTS. Including a discussion on why MCTS is preferred over other algorithms (e.g., 2-opt) and analyzing its synergy with the heatmap approach could add depth.
>
> For our stance on the novelty of our work, we kindly direct you to the  "**Opening Statement**" and the detailed discussion in the first point of  "Common Concerns" in the Global Response: "1. Regarding Novelty and  Research Values".
>
> Regarding the Comparison between MCTS and Other Search Algorithms, we respectfully disagree with the suggestion to include comparisons with other search algorithms for the following reasons:
>
> 1. As indicated by our paper title, our work specifically focuses on  analyzing the "Heatmap + MCTS" paradigm and the relative importance of  its components. Comparing MCTS with other search algorithms falls  outside the scope of our research objectives.
>
> 2. The superiority of MCTS-based approaches over other search algorithms  (such as beam search) for large-scale TSP has been well-established in  previous works. Given that the "Heatmap + MCTS" framework represents the  current state-of-the-art in solving large-scale TSP, we believe a  thorough analysis of this specific framework is both timely and  necessary.
>
> 3. Regarding the 2-opt algorithm mentioned in your comments, we have  already incorporated a more powerful k-opt operator in our MCTS  implementation, as described in Section 3.2 under "Simulation." The  k-opt operator inherently subsumes and outperforms the 2-opt algorithm,  making additional comparisons redundant.
>
>
>
> > W3: **Complexity and Sensitivity of MCTS Hyperparameters**: MCTS requires numerous hyperparameters and appears highly sensitive to these settings, which could make practical implementation challenging and limit accessibility.
>
> Given MCTS's remarkable success in solving TSP, despite its apparent  complexity, understanding its internal mechanics is crucial. Our work  systematically deconstructs the MCTS algorithm for TSP solving, and  Section 4.1 provides a detailed examination of the five most critical  parameters, effectively demystifying this seemingly black-box approach.
>
> Furthermore, our SHAP analysis precisely identifies the most influential  parameters, and we demonstrate that these can be efficiently optimized  using SMAC3 with minimal computational overhead. Rather than viewing  parameter tuning as a limitation, our work makes MCTS optimization more  accessible by providing clear, practical implementation guidelines.

---

> ### Author Response · Authors · 2024-11-19
> **Response to Reviewer DdHK (Part 2 of 2)**
>
> > Q1: **Generalization to Other CO Tasks**: The paper focuses on TSP. Do the authors anticipate that similar results could be achieved in other combinatorial optimization (CO) tasks, such as CVRP, or are these findings specific to TSP?
>
> Please refer to our response to W1.
>
>
>
> > Q2: **Clarification of Experimental Setup in Section 5.3**: In the "GENERALIZATION ABILITY" section, the experimental setup needs clarification. For example, for DIMES on TSP-1000, does the original (ORI.) setup use a model and MCTS tuned for TSP-500 or TSP-1000? Similarly, does the generalization (GEN.) setting use MCTS tuned specifically for TSP-1000 or for TSP-500? Adding a table or flowchart to illustrate the configurations would enhance clarity.
>
> We apologize for any confusion regarding Table 4. Let us clarify the experimental setup in detail:
> The table demonstrates the generalization capabilities of different methods from TSP500 to larger instances (TSP1k and TSP10k). Specifically:
>
> For "Gen." results:
>
> * DIMES, UTSP, and DIFUSCO: Models trained on TSP500 are used to generate heatmaps for TSP1k and TSP10k instances
>
> * SoftDist: Parameters tuned on TSP500 are used to generate heatmaps for larger instances
>
> * GT-Prior: The same heatmap obtained from TSP500 is directly used (requiring no additional inference)
>
> * All methods use MCTS parameters that were optimally tuned for each specific test scale
>
> For "Ori." results:
>
> * These correspond to Table 3, where models were trained/tuned specifically for each problem scale
>
> * We acknowledge a discrepancy between Tables 3 and 4/6 due to additional iterations in Table 3's results. We have corrected them in the updated manuscript.
>
> We appreciate your attention to detail and will enhance the clarity of this experimental setup description in the next version of the manuscript.

---

> > ### Comment · Reviewer_DdHK · 2024-11-27
> >
> > Thank you for answering my questions in detail.
> > I appreciate the effort you made to address my concerns and explain your work. While your responses have clarified some of my questions, I still feel that the paper, in its current state, is not yet ready for publication.
> > My remaining concerns are as follows:
> >
> > 1, I’m unsure of the value of solving just one specific TSP problem well. The scalability of the proposed algorithm is not apparent at this point.
> >
> > 2, While the paper provides a detailed discussion on MCTS and offers a helpful guide on its effective use, I believe it should also have offered an intuition for "why" MCTS fundamentally works well for TSP compared to other methods. However, this aspect is lacking in the current paper.
> >
> > After careful consideration, I have decided to maintain my original score.

---

> > > ### Author Response · Authors · 2024-11-27
> > >
> > > Thank you for your response.
> > > - First, we kindly remind you of the title of our paper,
> > >     > Rethinking the "Heatmap + Monte Carlo Tree Search'' Paradigm ***for Solving Large Scale TSP***.
> > >
> > >     Extending the heatmap+MCTS framework to other types of combinatorial  optimization problems falls outside the scope of our work. The primary focus of this paper is not on generalizing the heatmap+MCTS paradigm to other problems, but rather on addressing critical, previously overlooked  issues within the heatmap+MCTS  framework *specifically for tackling large-scale TSP*.
> > >
> > > - From an algorithmic design perspective, MCTS applied to TSP relies on the **k-opt operator** ,  a well-established local search method successfully applied in renowned TSP-solving algorithms such as LKH. The heatmap+MCTS framework leverages the heatmap's prior information to efficiently execute k-opt  operations, thereby improving computational efficiency. In addition, the balance between exploration and exploitation inherent to MCTS allows the framework to effectively escape local optima.
> > > From a performance perspective, the success of the heatmap+MCTS framework has been well-demonstrated in prior works[1,2,3]. Moreover, in the context of black-box optimization ,  hyperparameter tuning methods, such as Bayesian optimization, is a well-studied and widely recognized approach for improving solution performance. However, previous works have entirely overlooked the importance of hyper-parameter tuning for MCTS.
> > >
> > > From your response, it appears that there is a recurring focus on aspects that are either tangential to or beyond the intended scope of our work. This, unfortunately, diverts attention from the central messages and contributions of our paper. We sincerely hope that you can reassess the appropriate extent to which a single piece of work can be expected to cover and  re-evaluate our contributions in this context.
> > >
> > > The Authors.
> > >
> > >
> > > [1] Zhang-Hua Fu, Kai-Bin Qiu, and Hongyuan Zha. Generalize a small  pre-trained model to arbitrarily large tsp instances. In Proceedings of  the AAAI conference on artificial intelligence, volume 35, pp.  7474–7482, 2021.
> > >
> > > [2] Ruizhong Qiu, Zhiqing Sun, and Yiming Yang. Dimes: A  differentiable meta solver for combinatorial optimization problems.  Advances in Neural Information Processing Systems, 35:25531–25546, 2022.
> > >
> > > [3] Zhiqing Sun and Yiming Yang. Difusco: Graph-based diffusion  solvers for combinatorial optimization. Advances in Neural Information  Processing Systems, 36:3706–3731, 2023.

---

### Official Review · Reviewer_RLaV · 2024-11-04

**Soundness:** 3
**Presentation:** 3
**Contribution:** 2
**Rating:** 3
**Confidence:** 4

**Summary:**

This paper suggests a novel approach to solving the traveling salesman problem (TSP) by leveraging neural combinatorial optimization techniques, focusing on Monte-Carlo tree search (MCTS) that utilizes a probability distribution in the form of a heatmap for city-to-city edges. Although existing studies have also explored methods combining heatmaps and MCTS, this paper highlights that optimizing MCTS parameters within the conventional 'heatmap + MCTS' framework can enhance performance. Building on this insight, the authors propose a new, parameter-free algorithm for heatmap generation called GT-Prior. Experimental results from TSP-500, TSP-1000, and TSP-10000 demonstrate that performance improvements can be achieved through MCTS parameter optimization in current heatmap-based methods. Furthermore, the GT-Prior algorithm, when integrated with MCTS, achieves performance that is comparable to or slightly better than traditional approaches.

**Strengths:**

The paper is well-written and easy to follow. It provides a detailed explanation of MCTS algorithm parameter tuning for the TSP problem, dedicating significant effort to the experimental performance improvement and analysis of existing heatmap algorithms through MCTS parameter optimization. The newly proposed GT-Prior algorithm effectively captures the characteristics of the TSP problem and demonstrates a simple yet competitive performance compared to traditional heatmap generation algorithms.

**Weaknesses:**

As the authors mentioned, heatmap-based MCTS algorithms have been studied in the past. It would have been more impactful if the proposed MCTS algorithm in this paper had shown new technical contributions beyond parameter tuning. Although the study included an analysis of parameter importance using SHAP, it would have been more informative if the paper had elaborated on how these results were applied and their overall significance. Additionally, it is a limitation that the experiments were conducted solely within the TSP domain. It would have been more compelling if additional experiments had been performed across various CO domains, such as the Maximal Independent Set (MIS). Furthermore, while the paper addresses large-scale problems like TSP-1000 and TSP-10000, including real-world experiments using datasets like TSPLIB would have strengthened the work.

**Questions:**

1. Previous MCTS studies have employed parameter tuning methods like grid search. What fundamentally differentiates the parameter tuning approach in this paper from those in prior research?

2. Would it be possible for GT-Prior to be extended and applied to other combinatorial optimization domains, such as CVRP or MIS, beyond the TSP problem?

---

> ### Author Response · Authors · 2024-11-19
> **Response to Reviewer RLaV (Part 1 of 2)**
>
> Dear Reviewer RLaV:
>
> ### Opening Statement
>
> First of all, We earnestly request that you reconsider our work from a different  perspective. Our primary contribution does not lie in proposing a novel  hyperparameter tuning method; rather, we present a critical viewpoint  that challenges the current research trajectory in this field. Our work  aims to provoke a fundamental reassessment of the "Heatmap + MCTS"  paradigm for large-scale TSP solving. We firmly believe that our  insights and conclusions make a significant contribution by encouraging  the research community to critically examine established assumptions and  potentially redirect research efforts more effectively.
>
> > W1: **Lack of Novel Contributions**: Limited to parameter tuning without introducing new technical contributions.
>
> For our stance on the novelty of our work, we kindly direct you to the  "**Opening Statement**" and the detailed discussion in the first point of  "Common Concerns" in the Global Response: "1. Regarding Novelty and  Research Values".
>
>
> > W2: **Unclear Parameter Importance Application**: Although the study included an analysis of parameter importance using SHAP, it would have been more informative if the paper had elaborated on how these results were applied and their overall significance.
>
> We believe the SHAP analysis effectively reveals the contribution of each parameter to solution quality and have provided detailed instructions on how to interpret the SHAP plots in the caption of Fig. 1. Specifically, it serves two crucial  purposes:
>
> 1. It quantitatively demonstrates which parameters most impact performance, challenging previous works' default configurations.
>
> 2. It guides efficient tuning by identifying key parameters to focus on.
>
> Our performance improvements (Table 2) and efficient tuning results (Section 6) directly leverage these insights, validating the practical value of this analysis.
>
> > W3: **Experiments Limited to TSP**: Results would be more compelling with experiments in other domains, e.g., MIS.
>
> Please refer to the second point under "Common Concerns" in the Global Response: "2. Generalization to Other CO Tasks"
>
>
> > W4: **No Real-World Dataset Experiments**: Including tests on real-world datasets like TSPLIB would strengthen the work.
>
> We have conducted extensive experiments on the TSPLIB dataset, analyzing three categories:
>
> * Small (0-500 nodes, 43 instances)
>
> * Medium (500-2000 nodes, 21 instances)
>
> * Large (2000-18512 nodes, 14 instances)
>
> For detailed per-instance results, please refer to Table 7-12, Appendix E in the revised paper.
>
> Our experiments used two hyperparameter settings:
>
> 1. Tuned parameters optimized on uniform TSP instances
>
> 2. Default parameters from [1]
>
> |  |  | Zero | Att-GCN | DIMES | UTSP | SoftDist | DIFUSCO | GT-Prior |
> |---|---|---|---|---|---|---|---|---|
> | Small | Tuned | 0.79% | 0.67% | *0.48%* | **0.45%** | 1.23% | 0.58% | 0.76% |
> |  | Default | 0.87% | *0.67%* | 16.01% | 7.16% | 1.80% | 1.06% | **0.20%** |
> | Medium | Tuned | 15.24% | 11.47% | 10.64% | 12.03% | *6.79%* | **2.38%** | 10.08% |
> |  | Default | 4.88% | 3.73% | 4.06% | 13.58% | 7.20% | **2.87%** | *3.63%* |
> | Large | Tuned | 5.54% | *3.92%* | 5.48% | 26.51% | 4.43% | 5.68% | **3.52%** |
> |  | Default | 6.51% | **4.84%** | 391.89% | 1481.66% | 11.36% | 12.62% | *5.51%* |
>
> Our TSPLIB experiments reveal several key findings:
>
> 1. **Impact of Parameter Tuning**: The dramatic performance variations between tuned and default parameters (e.g., UTSP's 26.51% vs 1481.66% on large instances) demonstrate the substantial impact of hyperparameter tuning.
>
> 2. **Generalization Challenge**: The limited transferability of parameters from uniform instances to TSPLIB reveals a fundamental generalization issue shared by all methods except the Zero heatmap, further emphasizing the crucial role of MCTS tuning in the framework. To illustrate the distribution shift between uniform and TSPLIB instances, we present visualizations of representative hard and easy instances across different scales in Figure 6, Appendix E in the revised paper.
>
> 3. **Zero Heatmap Stability**: The Zero heatmap's consistently stable performance across various instance sizes and distributions validates our claim that MCTS's contribution has been historically undervalued.
>
> 4. **GT-Prior Effectiveness**: Our proposed GT-Prior demonstrates remarkable adaptability, achieving superior performance particularly on large instances (3.52% with tuned parameters) and maintaining consistent performance across different parameter settings, suggesting it effectively combines the benefits of TSP prior knowledge with MCTS optimization.
>
> 5. **Computational Efficiency**: Unlike learning-based baselines that heavily rely on GPUs for both training and inference, GT-Prior's independence from GPU requirements removes a significant bottleneck when dealing with large scale real-world instances.

---

> ### Author Response · Authors · 2024-11-19
> **Response to Reviewer RLaV (Part 2 of 2)**
>
> > Q1: **Parameter Tuning Difference**: How does the parameter tuning approach in this paper fundamentally differ from prior methods like grid search?
>
> No. However, as stated in our discussion on novelty and contribution, our primary objective is not to propose a new method, but rather to present a different perspective and validate it using straightforward approaches. Specifically, we employed grid search on MCTS parameters to obtain the best performance for each method type (which could only be definitively demonstrated through grid search) to illustrate two key points:
>
> 1. The critical importance of MCTS parameter tuning, as evidenced in Table 2: Performance Improvement after Hyperparameter Tuning.
>
> 2. The actual role of heatmaps within this framework, demonstrating that competitive performance can be achieved using a parameter-free, generalizable GT-Prior without any resource intensive model training.
>
> For practical applications, various advanced hyperparameter tuning methods can be employed. As shown in Appendix F, using SMAC3 for parameter tuning demonstrates that the entire optimization process can be completed efficiently.
>
>
> > Q2: **GT-Prior Extensibility**: Can GT-Prior be applied to other combinatorial optimization domains beyond TSP, such as CVRP or MIS?
>
> No, the non-generalizability of GT-Prior to other Combinatorial Optimization Problems (COPs) stems from two fundamental aspects:
>
> 1. Problem-Specific Structure: GT-Prior relies on statistical patterns derived from k-nearest neighbor relationships in optimal solutions, which is inherently tied to TSP's unique problem structure. This statistical prior is not naturally transferable to other COPs with different underlying structures.
>
> 2. TSP-Specific Search Operations: The MCTS implementation heavily depends on k-opt, a local search algorithm specifically designed for TSP.
>
> Therefore, to extend the "heatmap + MCTS" framework to other COPs, two problem-specific adaptations are essential:
>
> 1. Design problem-specific GT-Prior that captures the unique structural properties of the target problem
>
> 2. Embed problem-specific local search algorithms into the MCTS framework,
>
> We believe that these adaptations to other COPs, while interesting, fall beyond the scope of our current research objectives.
>
> Ref:
>
> [1] Zhang-Hua Fu, Kai-Bin Qiu, and Hongyuan Zha. Generalize a small pre-trained model to arbitrarily large tsp instances. In Proceedings of the AAAI conference on artificial intelligence, volume 35, pp. 7474–7482, 2021.

---

### Author Response · Authors · 2024-11-14
**Global Response**

We sincerely thank all SACs, PCs, ACs, and reviewers for their time and effort on our manuscript. While we appreciate the constructive comments, we believe there are some misunderstandings about our work's core contributions and significance. In this response, we will first highlight our key contributions to learning-based TSP solving, followed by addressing the common concerns raised by reviewers.

## Motivation of the "heatmap + MCTS" paradigm
The "heatmap + MCTS" paradigm first introduced by [1] and subsequently adopted by several state-of-the-art methods  [2,3,4], has emerged as one of the most successful approaches for solving large-scale TSP  instances with up to 10,000 nodes, significantly outperforming previous neural approaches. One should note, the main motivation of this series of works is:

**Main motivation**: To better solve large-scale TSPs

This motivation inspires an abundance of following works to develop more sophisticated heatmap generation methods seeking to push the boundary of solution quality. Here, please remember the **main motivation of all this line of work**.

## Our Motivation
We want to emphasize that our motivation aligns with the **Main motivation** stated above in the first place. Designing fancy heatmap generation methods, in our opinion, serves as a path to the **Main motivation**. Despite its remarkable success of "heatmap+MCTS", there has been a notable gap in systematically analyzing which component plays the pivotal role in this framework. Previous works have implicitly assumed the dominance of the heatmap component, directing most research efforts towards developing sophisticated learning-based heatmap construction methods. However, our comprehensive experiments reveal a surprising and novel finding: even with a simple zero-initialized heatmap devoid of any prior information, properly tuned MCTS parameters can achieve remarkably competitive results. This finding fundamentally challenges the prevailing assumption about the heatmap's primacy in the framework, and can be a big step towards the **Main motivation**.

## Our Contribution
Our work makes two significant contributions to address this research gap.
1. First, through systematic parameter tuning - the most straightforward approach - we demonstrate the crucial importance of MCTS  in the framework, which has been previously underestimated.
2. Second, we propose a parameter-free heatmap based on optimal solution priors, which exhibits strong generalization capabilities and achieves comparable  performance to many existing learning-based heatmaps.

While we acknowledge the impressive performance of previously proposed heatmaps, our findings raise important questions for the research community:

> **Should we continue investing substantial computational resources and research efforts in designing increasingly complex heatmaps for TSP?  Have we perhaps strayed from our primary goal of solving the TSP efficiently?**

We believe these questions merit serious consideration as we move forward in this field. And we believe these helpful findings represent substantial contributions, and stick to the **Main motivation** to better solve large-scale TSP, rather than proposing complicated methodical designs.

## Response to Common Concerns
### 1. Regarding Novelty and Research Values
We appreciate the reviewers' concerns about novelty, but we believe there might be a misunderstanding about the fundamental value and contribution of our work. We would like to emphasize that true scientific advancement doesn't always necessitate complex, novel methodologies; sometimes, **it requires challenging established assumptions and redirecting research focus through straightforward yet insightful analysis**.
Our work's novelty lies in its critical examination of a widely accepted paradigm rather than proposing another sophisticated solution. We deliberately chose simple, interpretable approaches to demonstrate our findings because they most clearly illuminate the overlooked aspects of the "heatmap + MCTS" framework.
We respectfully ask reviewers to reconsider our work's value through this lens as a contribution that **promotes critical thinking** and **potentially redirects research efforts** toward more efficient problem-solving approaches.
### 2. Generalization to Other CO Tasks
As proposed in the title of our manuscript, we specifically focused on TSP as the "Heatmap + MCTS" paradigm is uniquely successful in this domain, with state-of-the-art results from multiple recent works [1,2,3]. Notably, this paradigm has not yet been successfully adapted to other CO problems - current leading approaches for CVRP and other tasks use different methodologies entirely.

We will address the specific concerns raised by each reviewer in the following individual responses. We sincerely appreciate your time and  attention in reading through this global response.

The Authors

---

> ### Author Response · Authors · 2024-11-14
> **References (Updating)**
>
> [1] Zhang-Hua Fu, Kai-Bin Qiu, and Hongyuan Zha. Generalize a small pre-trained model to arbitrarily large tsp instances. In Proceedings of the AAAI conference on artificial intelligence, volume 35, pp. 7474–7482, 2021.
>
> [2] Ruizhong Qiu, Zhiqing Sun, and Yiming Yang. Dimes: A differentiable meta solver for combinatorial optimization problems. Advances in Neural Information Processing Systems, 35:25531–25546, 2022.
>
> [3] Zhiqing Sun and Yiming Yang. Difusco: Graph-based diffusion solvers for combinatorial optimization. Advances in Neural Information Processing Systems, 36:3706–3731, 2023.
>
> [4] Yimeng Min, Yiwei Bai, and Carla P Gomes. Unsupervised learning for solving the travelling salesman problem. Advances in Neural Information Processing Systems, 36, 2024.

---

### Meta-Review · Area_Chair_gyV7 · 2024-12-28

**Metareview:**

This work re-evaluates the “heat map + MCTS” paradigm for solving large traveling salesmen problems (TSP) (e.g., those with up to 10,000 nodes).  This work has the potential to change the way researchers think about large-scale TSP. The authors run extensive experiments examining the effect of MCTS HPs, and propose simple baselines (“GT-Prior” or “Zero”)  that when combined with HP optimization result in optimization performance that rivals competing approaches.

Many reviewers found the scope of the paper to be narrow (focused exclusively on TSP) (RLaV, DdHK), and did not find the contribution of highlighting the importance of HPs for heat map algorithms to be sufficiently compelling for publication at ICLR.  All reviewers did not advocate for acceptance, even after the rebuttal period and reviewer discussion.  In general, this paper is a bit of an odd fit for a conference on representation learning, and a different audience, such as AAAI or perhaps NeurIPS may have an audience that can better appreciate and critique the contributions.

As an aside: it would be interesting to use more standard methods for measuring sensitivity to HPs, such as Sobol indices or derivative-based methods for hyper parameter importance.  Some findings about what ranges of HPs tend to work best across problems could be helpful.  If importances do not generalize across problems, that could be put more simply in the main text.

**Additional Comments On Reviewer Discussion:**

The authors clarified many of the points of confusion, argued for the contribution of their work to understanding the performance of heatmap+MCTS algorithms (vs technical novelty) and its impact on TSP.  They ran additional experiments showing good results on TSPLIB.  However, reviewers were not swayed by these arguments, finding the technical contribution and impact to be too narrow.

---

### Decision · Program_Chairs · 2025-01-22

Reject